# From Language Models over Tokens to Language Models over Characters

**Tim Vieira** [1]   **Benjamin LeBrun** [2]   **Mario Giulianelli** [1]   **Juan Luis Gastaldi** [1]   **Brian DuSell** [1]   **John Terilla** [3]
**Timothy J. O'Donnell** [4 2 5]   **Ryan Cotterell** [1]

## Abstract

Modern language models are internally—and mathematically—distributions over *token* strings rather than *character* strings, posing numerous challenges for programmers building user applications on top of them. For example, if a prompt is specified as a character string, it must be tokenized before passing it to the token-level language model. Thus, the tokenizer and consequent processing are very sensitive to the specification of the prompt (e.g., whether the prompt ends with a space or not). This paper presents algorithms for converting token-level language models to character-level ones. We present both exact and approximate algorithms. In the empirical portion of the paper, we benchmark the practical runtime and approximation quality. Across four publicly available language models, we find that—even with a small computation budget—our method is able to accurately approximate the character-level distribution at reasonably fast speeds, and that a significant improvement in the language model's compression rate (bits/byte) is achieved.

 https://github.com/genlm/genlm-bytes

## 1. Introduction

Modern language models are engineered as probability distributions over strings of *tokens* rather than strings of *characters*. However, this leads to a fundamental tension between the users of language models and the engineers who build them. Specifically, token-level models are rife with unintuitive behaviors that—without a technical fix—baffle users. As an illustrative example of a common user complaint, we exhibit the prompt boundary problem (see below). This paper provides a principled solution to the prompt boundary problem as well as other oddities that make interfacing with token-level language models with character-level prompts hard for users.

**Tokenized language models: A brief overview.**[1]   Let $\Sigma$ be an alphabet of characters, and let $\Sigma^*$ denote the set of all strings that can be built from it. Suppose there is a true distribution $p_\Sigma^*$ over $\Sigma^*$ that we seek to model. We observe a training corpus of character strings: $\boldsymbol{\sigma}^{(1)}, \dots, \boldsymbol{\sigma}^{(M)} \overset{\text{i.i.d.}}{\sim} p_\Sigma^*$. However, rather than estimating a language model that approximates $p_\Sigma^*$ directly, we employ a (possibly stochastic) tokenizer $\tau$ that transforms the training corpus into a corpus of *token* strings. $\boldsymbol{\delta}^{(1)} \sim \tau(\cdot \mid \boldsymbol{\sigma}^{(1)}), \dots, \boldsymbol{\delta}^{(M)} \sim \tau(\cdot \mid \boldsymbol{\sigma}^{(M)})$. Next, we estimate a token-level language model to fit the strings $\boldsymbol{\delta}^{(1)}, \dots, \boldsymbol{\delta}^{(M)}$. Lastly, we use $p_\Delta$ to generate character strings through the following generative process: (i) sample $\boldsymbol{\delta} \sim p_\Delta$ and (ii) return $\boldsymbol{\sigma} = \kappa(\boldsymbol{\delta})$ where $\kappa$ is a decoding function. Let $p_\Sigma$ denote the resulting distribution of this process. Practically, we hope that the choice of $\tau$ and $\kappa$ should aid in our ability to estimate $p_\Sigma^*$ in the sense that $p_\Sigma^* \approx p_\Sigma$. Commonly used tokenizers in the realm of LLMs use tokenizers $\tau$ that break long strings into chunks. Intuitively, generating chunks instead of individual characters helps because it effectively shortens strings without obfuscating them using a complicated encoding.

**The prompt boundary problem.**   Consider the case of GPT2 (Radford et al., 2019), which was trained over token strings created from byte-pair encoding (BPE; Sennrich et al. (2016); Gage (1994)). Suppose we wish to generate continuations of the prompt: `"In␣the␣kingdom␣of␣the␣blind,␣the`. Unfortunately, the *interface* to $p_\Delta$ is not ideal: it does not accept a *character* string; thus, it is common to encode it as a *token* string with an encoding function $\tau_{\text{BPE}}$:[2,3]

$\tau_{\text{BPE}}(\texttt{"In␣the␣kingdom␣of␣the␣blind,␣the})$

$= \big[\underset{1}{\texttt{"}}, \underset{818}{\texttt{In}}, \underset{262}{\texttt{␣the}}, \underset{13239}{\texttt{␣kingdom}}, \underset{286}{\texttt{␣of}}, \underset{262}{\texttt{␣the}}, \underset{7770}{\texttt{␣blind}}, \underset{11}{\texttt{,}}, \underset{262}{\texttt{␣the}}\big]$

---

[1]ETH Zürich [2]Mila [3]City University of New York  [4]McGill University  [5]Canada CIFAR AI Chair . Correspondence to: Tim Vieira <tim.f.vieira@gmail.com>.

*Proceedings of the 42$^{nd}$ International Conference on Machine Learning*, Vancouver, Canada. PMLR 267, 2025. Copyright 2025 by the author(s).

---

[1]We adopt the notation of Gastaldi et al. (2025), who gave a general characterization for when training with tokenization allows for consistent estimation of the true distribution over characters.

[2]In practice, $\tau_{\text{BPE}}$ outputs an integer sequence; we provide the substring `gloss` for readability.

[3]We write $\boldsymbol{\delta} = \tau(\boldsymbol{\sigma})$ when $\tau$ is deterministic (i.e., a function).

If we complete the prompt by taking the most likely next token (*greedy completion*), we generate the following:

$$\begin{bmatrix} \text{\_one} & - & \text{eyed} & \text{\_man} & \text{\_is} & \text{\_king} & ." \\ 530 & 12 & 18834 & 582 & 318 & 5822 & 526 \end{bmatrix}$$

Now that we have our generated output, we can apply the decoding function $\kappa$ that maps token strings to character strings as part of the tokenization protocol:

$$\kappa_{\mathrm{BPE}}(\cdots) = \text{"In\_the\_kingdom\_of\_the\_blind,\_the}$$
$$\text{\_one-eyed\_man\_is\_king."}$$

This is a good completion, as the string is a well-known proverb. However, if we tweak the prompt *ever so slightly* by inserting a trailing whitespace:

$$\tau_{\mathrm{BPE}}(\text{"In\_the\_kingdom\_of\_the\_blind,\_the\_})$$
$$= \begin{bmatrix} " & \text{In} & \text{\_the} & \text{\_kingdom} & \text{\_of} & \text{\_the} & \text{\_blind} & , & \text{\_the} & \text{\_} \\ 1 & 818 & 262 & 13239 & 286 & 262 & 7770 & 11 & 262 & 220 \end{bmatrix}$$

greedy completion gives $\begin{bmatrix} \text{ills} & \text{\_of} & \text{\_the} & \text{\_world} & \text{\_are} & \text{\_seen} & \cdots \\ 2171 & 286 & 262 & 995 & 389 & 1775 \end{bmatrix}$ because the conditional probability $(\overrightarrow{p_\Delta})$ of generating the characters we want becomes highly unlikely, dropping from $0.98 = \overrightarrow{p_\Delta}(\text{\_one}_{530} \mid \tau_{\mathrm{BPE}}(\cdots \text{\_the}))$ to $5 \times 10^{-7} = \overrightarrow{p_\Delta}(\text{one}_{505} \mid \tau_{\mathrm{BPE}}(\cdots \text{\_the\_}))$ upon appending the $\text{\_}$ character. Hence, the trailing white space undesirably impacts the output, which no longer corresponds to the proverb.

We formalize the **prompt boundary problem** as follows: Let $\sigma \in \Sigma^*$ be our initial prompt, such as "In\_the\_kingdom\_of\_the\_blind,\_the). Now, consider a pair of future strings $\sigma' \cdot \alpha$ and $\sigma' \cdot \beta$ with a common prefix $\sigma'$ (e.g., \_one and \_ills) We say a language model interface suffers from the prompt boundary problem if moving the common prefix $\sigma'$ (e.g., \_) across the boundary into the prompt does not preserve the relative probability of the pair of future strings, i.e., the relative probability of $\sigma' \cdot \alpha$ and $\sigma' \cdot \beta$ given $\sigma$ does not equal the relative probability of $\alpha$ and $\beta$ given $\sigma \cdot \sigma'$. Our example fails this test because the probability of \_one was much higher than \_ills, but then, the relative order swaps after moving \_ into the prompt.

Our perspective is that the prompt boundary problem arises from incorrectly conditioning the *token-level* language model on a string of *characters* by using $\tau(\sigma)$ rather than finding token strings that best match the prompt $\sigma$.[4]

$\hookrightarrow$ *The token healing heuristic:* Lundberg & Ribeiro (2023) present *token healing* as a heuristic that mitigates the prompt boundary problem; it works as follows: (1) Tokenize (i.e., encode) the prompt. (2) Backup the tokenized prompt to the penultimate token. (3) Generate the next token subject to the constraint that it starts with the unmatched substring at the

end of the prompt. (4) Continue generating as usual. Now, we can see how token healing patches the running example:

$$\tau_{\mathrm{BPE}}(\text{"In\_the\_kingdom\_of\_the\_blind,\_the\_})$$
$$= \begin{bmatrix} " & \text{In} & \text{\_the} & \text{\_kingdom} & \text{\_of} & \text{\_the} & \text{\_blind} & , & \text{\_the} & \text{\textbackslash} \\ 1 & 818 & 262 & 13239 & 286 & 262 & 7770 & 11 & 262 & 220 \end{bmatrix}$$

The most probable next token starting with \_ is $\text{\_one}_{530}$. Now, generating the remaining tokens recovers the desired output, as it matches the prompt before we added the whitespace.

Unfortunately, backing up one token is insufficient for the general case, as the following example will illustrate. Consider generating from GPT2 using the prompt Hello,\_worl:

$$\tau_{\mathrm{BPE}}(\text{Hello,\_worl}) = \begin{bmatrix} \text{Hello} & , & \text{\_wor} & \text{l} \\ 15496 & 11 & 476 & 75 \end{bmatrix}$$

The most likely next *character* ought to be d as Hello,\_world is a common expression (popularized in educational material). However, the most likely *token* results in Hello,\_worlwide, an apparent misspelling of worldwide.[5] Unfortunately, token healing's strategy of backing up by a single token cannot salvage the poor tokenization as $\text{l}_{75}$ is still the most common next token that is consistent with the string l. Thus, after generating $\text{l}_{75}$, we are back where we started, generating $\begin{bmatrix} \text{Hello} & , & \text{\_wor} & \text{l} & \text{wide} \\ 15496 & 11 & 476 & 75 & 4421 \end{bmatrix}$.

$\hookrightarrow$ *Getting it right:* A simple "probability 101" expression tells us the correct solution to the prompt boundary problem. Consider a $\Delta^*$-valued random variable $Y$, distributed according to $p_\Delta$. Then, the correct way to sample from $p_\Delta$ conditioned on a character string $\sigma$ is according to

$$p_{\Delta|\Sigma}(\delta \mid \sigma) \stackrel{\text{def}}{=} \mathop{\mathbb{P}}_{Y \sim p_\Delta}[Y = \delta \mid \kappa(Y) \succeq \sigma] \qquad (1)$$

where we have conditioned on the event that the decoded string $\kappa(Y)$ has $\sigma$ as a prefix (i.e., $\kappa(Y) \succeq \sigma$). While innovative with respect to the literature, the expression $\mathbb{P}_{Y \sim p_\Delta}[Y = \delta \mid \kappa(Y) \succeq \sigma]$ conveys the probability we are interested in precisely and concisely. For the more procedurally minded, this corresponds to the following process:

```
1 def conditional_token_generation(σ):
2   while True:
3     sample δ ∼ p_Δ
4     if κ(δ) ⪰ σ: return δ # accept
```

This is, of course, very inefficient, but fear not—we will provide an equivalent, efficient algorithm.

Our method finds a set of token strings that form a *covering*, a key technical concept we introduce in this paper. We will provide the precise definition in due course; for now, we will illustrate the covering of Hello,\_worl:

---

[4] Probabilistic condition preserves the necessary relative probabilities that we specified in the prompt boundary problem.

[5] The misspelling is a testament to the extent to which the tokenized prompt is out-of-distribution.

$$\blacksquare (0.9820714) : \begin{bmatrix} \text{Hello} & , & \_\text{world} \\ 15496, & 11, & 995 \end{bmatrix}$$

$$| (0.0106702) : \begin{bmatrix} \text{Hello} & , & \_\text{worlds} \\ 15496, & 11, & 11621 \end{bmatrix}$$

$$| (0.0070749) : \begin{bmatrix} \text{Hello} & , & \_\text{worldwide} \\ 15496, & 11, & 8688 \end{bmatrix}$$

$$| (0.0000830) : \begin{bmatrix} \text{Hello} & , & \_\text{worldly} \\ 15496, & 11, & 43249 \end{bmatrix}$$

$$| (0.0000369) : \begin{bmatrix} \text{Hello} & , & \_\text{worldview} \\ 15496, & 11, & 29081 \end{bmatrix}$$

$$| (0.0000225) : \begin{bmatrix} \text{Hell} & \text{o} & , & \_\text{world} \\ 28254, & 78, & 11, & 995 \end{bmatrix}$$

$$| (0.0000179) : \begin{bmatrix} \text{Hello} & , & \_\text{wor} & \text{l} \\ 15496, & 11, & 476, & 75 \end{bmatrix}$$

$$\vdots$$

Notice that each token string in the covering has the character string `Hello,␣worl` as a prefix (after decoding), but it may have some extra characters (marked by underlining) in the partially matched last token—just like token healing. The size of the complete covering may, in general, be exponential in the length of the character string. However, we can enumerate its high-probability members very quickly in practice. Since the worst-case time to find the top-$K$ elements of the covering might be exponential, we provide a more aggressive approximation based on beam search that gives a good approximation even with small beam sizes.

Unlike token healing, the sequence $\begin{bmatrix} \text{Hello} & , & \_\text{world} \\ 15496, & 11, & 995 \end{bmatrix}$ is the highest probability member of the covering; thus, it is not pruned away by the aggressive heuristics based on $\tau$, as in token healing. The covering is defined only in terms of $\kappa$, and the pruning is based on the language model probability, which typically prioritizes the token strings that are most reflective of the language model's training data. We provide an algorithm for correctly conditioning a token-level model on a character string in §3.4.

**Character-level model.** At the character level, working with the language model is intuitive. Consider, again, our example illustrating the prompt boundary problem. Appending whitespace behaves predictably, as it satisfies the *probabilistic chain rule*:

$$\overrightarrow{p_\Sigma}(\_\text{one} \mid \text{"In␣the␣kingdom␣of␣the␣blind,␣the})$$
$$= \overrightarrow{p_\Sigma}(\_ \mid \text{"In␣the␣kingdom␣of␣the␣blind,␣the})$$
$$\cdot \overrightarrow{p_\Sigma}(\text{one} \mid \text{"In␣the␣kingdom␣of␣the␣blind,␣the␣})$$

Here, $\overrightarrow{p_\Sigma}$ denotes the character-level model's conditional distribution. Similarly, recall the `Hello,␣world` example above. The character-level model correctly infers that `d` is the most likely next character given `Hello,␣worl`. The computation of this conditional probability is simply the total probability of the covering of `Hello,␣world` divided by the total probability of the covering of `Hello,␣worl`. These quantities are derived from our concept of covering, which directly leads to an algorithm for determining the distribution over possi-

ble next characters. Beyond the prompt boundary problem, computing the conditional probability of a character string given a token-level language model has many applications.

**Applications.** Aside from making character-level conditioning well-behaved, we highlight a few applications of language models requiring careful character-level reasoning:

↪ *Character-level constraints:* Enforcing character-level constraints on allowed strings is a promising area that has received much recent attention (e.g., Scholak et al., 2021; Poesia et al., 2022; Geng et al., 2023; Microsoft, 2023; Willard & Louf, 2023; Koo et al., 2024; Loula et al., 2025).

↪ *Computational psycholinguistics:* Computing the contextual surprisal (negative log probability) of a character substring to predict human reading times (Hale, 2001; Levy, 2008). Two recent papers (Oh & Schuler, 2024; Pimentel & Meister, 2024) have given algorithms for computing the surprisal of whitespace-separated *words* under a number of strong assumptions. Our algorithms can compute the contextual surprisal of arbitrary character strings. Giulianelli et al. (2024) show experimentally that having the flexibility to compute character substring surprisals leads to a more predictive model of reading time than a fixed notion of a word.

**Does it work?** In the experimental portion of our paper (§4), we report the empirical runtime of our algorithm for converting token-level language models to character-level ones and quantify its accuracy in estimating the conditional distribution over characters. We find that even with a limited computational budget, our method provides an accurate estimate of the conditional distribution over the next character under four publicly available language models.[6] We also find that the compression rate (bits/byte) is significantly improved by estimating the probability of the corpus as a character string rather than a canonical token string.

## 2. Background

### 2.1. Alphabets and Strings

An **alphabet** $\Gamma$ is a non-empty, finite set of elements called **symbols**. A **string** $\gamma$ over alphabet $\Gamma$ is a finite sequence $\gamma = \gamma_1 \cdots \gamma_N$ for some $0 \le N < \infty$ of symbols where $\gamma_1, \dots, \gamma_N \in \Gamma$. Let $|\gamma|$ denote the string's length $N$. We denote the empty string as $\varepsilon$. For any alphabet $\Gamma$, let $\Gamma^*$ denote the set of all strings over $\Gamma$, and let $\Gamma^+$ denote the set of all non-empty strings over $\Gamma$. For any two strings $\gamma', \gamma'' \in \Gamma^*$, we denote their concatenation as $\gamma' \cdot \gamma''$. Additionally, we define $S \cdot S' \stackrel{\text{def}}{=} \{\gamma \cdot \gamma' \mid \gamma \in S, \gamma' \in S'\}$ for any $S, S' \subseteq \Gamma^*$. Given a string $\gamma$ such that $|\gamma| \ge t$, let $\gamma_{<t}$ denote the string made from the first $t-1$ characters of $\gamma$. We write $\gamma \preceq \gamma'$

---

[6]Specifically, Llama-3.2-1B, Meta-Llama-3.1-8B, DeepSeek-R1-Distill-Llama-8B, and phi-4 (14B).

if $\gamma$ is a prefix of $\gamma'$ and $\gamma \prec \gamma'$ if $\gamma$ is a proper prefix of $\gamma'$. The relation $\preceq$ defines a partial order on $\Gamma^*$. We write $\succeq$ and $\succ$ to refer to the relations $\preceq$ and $\prec$ with their respective arguments transposed.

## 2.2. Language Models and Prefix Probability

A **language model** $p_\Gamma$ is a probability distribution over $\Gamma^*$ where $\Gamma$ is an alphabet. Let $Y$ be a $\Gamma^*$-valued random variable distributed according to $p_\Gamma$ and $\gamma \in \Gamma^*$. The **prefix probability** $\overrightarrow{p_\Gamma}(\gamma)$ is the probability that $Y$ has prefix $\gamma$:

$$\overrightarrow{p_\Gamma}(\gamma) \stackrel{\text{def}}{=} \mathop{\mathbb{P}}_{Y \sim p_\Gamma} [Y \succeq \gamma] = \sum_{\gamma' \in \Gamma^*} \mathbb{1}\{\gamma' \succeq \gamma\} \, p_\Gamma(\gamma') \quad (2)$$

We also define the following shorthand for the **conditional prefix probability** $\overrightarrow{p_\Gamma}(\gamma' \mid \gamma)$ as the probability of the event $Y \succeq \gamma \cdot \gamma'$ provided that $Y \succeq \gamma$:

$$\overrightarrow{p_\Gamma}(\gamma' \mid \gamma) \stackrel{\text{def}}{=} \mathop{\mathbb{P}}_{Y \sim p_\Gamma} [Y \succeq \gamma \cdot \gamma' \mid Y \succeq \gamma] = \frac{\overrightarrow{p_\Gamma}(\gamma \cdot \gamma')}{\overrightarrow{p_\Gamma}(\gamma)} \quad (3)$$

We may express the probability of $\gamma$ as a product of conditional prefix probabilities:

$$p_\Gamma(\gamma) = \overrightarrow{p_\Gamma}(\text{EOS} \mid \gamma) \prod_{t=1}^{|\gamma|} \overrightarrow{p_\Gamma}(\gamma_t \mid \gamma_{<t}) \quad (4)$$

where each $\overrightarrow{p_\Gamma}(\gamma_t \mid \gamma_{<t})$ is an instance of Eq. (3), and

$$\overrightarrow{p_\Gamma}(\text{EOS} \mid \gamma) \stackrel{\text{def}}{=} \frac{p_\Gamma(\gamma)}{\overrightarrow{p_\Gamma}(\gamma)} \quad (5)$$

Here, EOS is a distinguished **end-of-string symbol** that cannot appear in any alphabet. Of particular interest are the single-symbol conditional prefix distributions $\overrightarrow{p_\Gamma}(\cdot \mid \gamma_{<t})$, as they may each be interpreted as a probability distribution over the set $\Gamma \cup \{\text{EOS}\}$—in fact, modern language models[7] are *defined* via the product in Eq. (4) where each single-symbol conditional prefix probability comes from the learned parametric model.[8]

## 2.3. Tokenization

We now discuss our basic formalization for tokenization.

**Definition 1.** *An (exact) tokenization model is a tuple* $(\Sigma, \Delta, \tau, \kappa)$ *where*

• $\Sigma$ *is an alphabet of **character** symbols*

• $\Delta$ *is an alphabet of **token** symbols*
• $\tau$ *is a (possibly) stochastic **encoder**:* $\tau(\cdot \mid \sigma)$ *is a probability distribution over* $\Delta^*$ *for each* $\sigma \in \Sigma^*$
• $\kappa \colon \Delta^* \to \Sigma^*$ *is a **decoder** function satisfying **exactness*** $\sum_{\delta \in \Delta^*} \mathbb{1}\{\kappa(\delta) = \sigma\} \tau(\delta \mid \sigma) = 1$, *for all* $\sigma \in \Sigma^*$

**Definition 2.** *A **tokenized language model** $p_\Sigma$ is a language model over $\Sigma^*$ that is parameterized by a language model $p_\Delta$ over $\Delta^*$ and a decoding function $\kappa \colon \Delta^* \to \Sigma^*$. This tokenized language model generates character strings via the following process: (i) $\delta \sim p_\Delta$, (ii) $\sigma \leftarrow \kappa(\delta)$. Thus, the character strings $\sigma$ generated have the distribution:*

$$p_\Sigma(\sigma) \stackrel{\text{def}}{=} \mathop{\mathbb{P}}_{Y \sim p_\Delta} [\kappa(Y) = \sigma] \quad (6)$$

Note that $p_\Sigma(\sigma)$ accounts for the fact that many token strings may be associated with a given character string through $\kappa$.[9] To describe that association, we define $\mathcal{E}(\sigma) \stackrel{\text{def}}{=} \{\delta \in \Delta^* \colon \sigma = \kappa(\delta)\}$, the set of **encodings** for any character string $\sigma \in \Sigma^*$.[10]

**What about $\tau$?** The reader may notice that $\tau$ does not appear in Eq. (6). Although $\tau$ is essential for generating training data, once the model $p_\Delta$ has been trained, the information in $\tau$ is not of immediate practical use. Moreover, attempts to leverage $\tau$ seem to lead to faulty heuristics, as we discussed in the introduction. We note that under exactness, $\tau(\sigma)$ must be present in $\mathcal{E}(\sigma)$. This is because exactness implies that $\mathcal{E}(\sigma) \supseteq \{\delta \in \Delta^* \colon \tau(\delta \mid \sigma) > 0\}$ for all $\sigma \in \Sigma^*$. In the common case where $\tau$ is **deterministic**[11] we emphasize that $\mathcal{E}(\sigma) \supseteq \{\tau(\sigma)\}$. The tokenization model would need to be **bijective**[12] for $\mathcal{E}(\sigma) = \{\tau(\sigma)\}$. Unfortunately, common tokenizers (e.g., BPE) are *not* bijective.

**The mirage of the canonical tokenization.** Consider the case when the encoder $\tau$ is deterministic. In that case, we write $\delta = \tau(\sigma)$, and we call this $\delta$ the **canonical** tokenization of $\sigma$. Note that even if $\tau$ is deterministic, there may exist many **noncanonical** tokenizations $\delta' \in \mathcal{E}(\sigma)$ such that $\delta' \neq \tau(\sigma)$ with nonzero probability $p_\Delta(\delta') > 0$. Thus, the character string generation process includes a mix of canonical and noncanonical token strings—making it incorrect to only consider a character string's canonical tokenization when assessing its probability. In practice, the conditional probability $\mathbb{P}_{Y \sim p_\Delta}[Y = \delta \mid \kappa(Y) = \sigma]$ over the encodings $\delta$ of a character string $\sigma$ tends to be highly

---

[7]E.g., transformers (Vaswani et al., 2017), RNNs (e.g., Mikolov et al., 2010), and $n$-gram models (e.g., Shannon, 1948).

[8]The reader may notice that the equations for $p_\Gamma(\gamma)$, $\overrightarrow{p_\Gamma}(\gamma' \mid \gamma)$, and $\overrightarrow{p_\Gamma}(\text{EOS} \mid \gamma)$ are mutually recursive. Thus, they may *appear* circular. The key to resolving this concern is to recognize that $p_\Gamma$ is given as a *base case*. We note, however, that some readers may view Eq. (4) as the *definition* of the language model $p_\Gamma$, taking the components on the right-hand side of Eq. (4) as the base case.

[9]Many authors (Cao & Rimell, 2021; Chirkova et al., 2023; Phan et al., 2024) have discussed this particular complication; however, no algorithms for computing the next-character distribution exist.

[10]Note that $|\mathcal{E}(\sigma)|$ can be very large, e.g., infinite in the worst case. In the case of BPE, it is exponential in $|\sigma|$.

[11]I.e., $\tau(\delta \mid \sigma) \in \{0, 1\}$ for all $\delta \in \Delta^*, \sigma \in \Sigma^*$.

[12]I.e., a bijective tokenizer is deterministic, exact, and satisfies $\tau(\kappa(\delta)) = \delta$ for all $\delta \in \Delta^*$.

concentrated on the canonical tokenizations, as illustrated in Example 1 below.

**Example 1.** *Below, we show* GPT2*'s top encodings from* $\mathcal{E}(\texttt{Hello,\_world})$*, ranked by their conditional probability.*

*This short string has 78 encodings from numerous partitions of* Hello,\_world *into substrings of the tokenization alphabet* $\Delta$*. We see that the probability heavily concentrates on the top string, which is canonical.*

$$\blacksquare(0.9999719) : \begin{bmatrix} \text{Hello} & , & \text{\_world} \\ 15496 & 11 & 995 \end{bmatrix}$$

$$|(0.0000229) : \begin{bmatrix} \text{Hell} & \text{o} & , & \text{\_world} \\ 28254 & 78 & 11 & 995 \end{bmatrix}$$

$$(0.0000024) : \begin{bmatrix} \text{Hello} & , & \text{\_wor} & \text{ld} \\ 15496 & 11 & 476 & 335 \end{bmatrix}$$

$$(0.0000017) : \begin{bmatrix} \text{He} & \text{llo} & , & \text{\_world} \\ 1544 & 18798 & 11 & 995 \end{bmatrix}$$

$$(0.0000004) : \begin{bmatrix} \text{H} & \text{ell} & \text{o} & , & \text{\_world} \\ 39 & 695 & 78 & 11 & 995 \end{bmatrix}$$

$$(0.0000002) : \begin{bmatrix} \text{Hello} & , & \text{\_w} & \text{orld} \\ 15496 & 11 & 266 & 1764 \end{bmatrix}$$

$$(0.0000002) : \begin{bmatrix} \text{H} & \text{ello} & , & \text{\_world} \\ 39 & 11109 & 11 & 995 \end{bmatrix}$$

**A character-level interface.** A character-level interface to the token-level language model $p_\Delta$ is available in the following equations, which hold $\forall \boldsymbol{\sigma}, \boldsymbol{\sigma}' \in \Sigma^*$:

$$\overrightarrow{p_\Sigma}(\boldsymbol{\sigma}) = \mathop{\mathbb{P}}_{Y \sim p_\Delta}[\kappa(Y) \succeq \boldsymbol{\sigma}] \tag{7}$$

$$\overrightarrow{p_\Sigma}(\boldsymbol{\sigma}' \mid \boldsymbol{\sigma}) = \frac{\overrightarrow{p_\Sigma}(\boldsymbol{\sigma} \cdot \boldsymbol{\sigma}')}{\overrightarrow{p_\Sigma}(\boldsymbol{\sigma})} \tag{8}$$

$$\overrightarrow{p_\Sigma}(\text{EOS} \mid \boldsymbol{\sigma}) = \frac{p_\Sigma(\boldsymbol{\sigma})}{\overrightarrow{p_\Sigma}(\boldsymbol{\sigma})} \tag{9}$$

These equations show that we can have a complete character-level language model derived from the tokenized language model if we can compute—or approximate—the necessary summations implied by Eq. (6) and (7); specifically,

$$p_\Sigma(\boldsymbol{\sigma}) = \sum_{\boldsymbol{\delta} \in \Delta^*} \mathbb{1}\{\kappa(\boldsymbol{\delta}) = \boldsymbol{\sigma}\} \, p_\Delta(\boldsymbol{\delta}) \tag{10}$$

$$\overrightarrow{p_\Sigma}(\boldsymbol{\sigma}) = \sum_{\boldsymbol{\delta} \in \Delta^*} \mathbb{1}\{\kappa(\boldsymbol{\delta}) \succeq \boldsymbol{\sigma}\} \, p_\Delta(\boldsymbol{\delta}) \tag{11}$$

We will develop effective methods for these summations for the family of strict-prefix monotone decoders $\kappa$ (described in §2.4) where Eq. (10) and Eq. (11) admit a finite summation.

### 2.4. Key Properties of $\kappa$

This section defines the essential properties of $\kappa$ we require.

**Definition 3.** *We say* $\kappa \colon \Delta^* \to \Sigma^*$ *is*
• *prefix monotone if* $\boldsymbol{\delta} \preceq \boldsymbol{\delta}' \implies \kappa(\boldsymbol{\delta}) \preceq \kappa(\boldsymbol{\delta}')$
• *strict-prefix monotone if* $\boldsymbol{\delta} \prec \boldsymbol{\delta}' \implies \kappa(\boldsymbol{\delta}) \prec \kappa(\boldsymbol{\delta}')$

In simpler terms, strict-prefix monotonicity implies that concatenating a token to the encoding necessarily concatenates at least one character to the decoded character string. The diagrams below illustrate how GPT2's strict-prefix monotone $\kappa$ gives rise to a certain alignment between three token strings and the character string Hello,\_world:

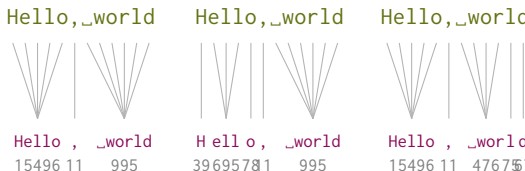

More formally, every application $\sigma_1 \cdots \sigma_M = \kappa(\delta_1 \cdots \delta_M)$ of a strict-prefix monotone mapping has the following properties. Each token in $\delta_1 \cdots \delta_M$ maps to one or more contiguous characters in $\sigma_1 \cdots \sigma_M$. Moreover, the mappings do not exclude any characters, and no edges of the mapping cross one another. *Strict* prefix monotonicity, in contrast to prefix monotonicity, ensures that there are no deletions of tokens in the mapping, i.e., each token maps to *at least one* character.

Strict-prefix monotonicity is the key structural property required by §3's algorithms, as it allows us to replace an infinite sum with a finite sum in Proposition 1. We briefly mention an important special case. A decoder $\kappa$ is **multiplicative** if $\kappa(\delta_1 \cdots \delta_N) = \kappa(\delta_1) \cdots \kappa(\delta_N)$ for all $\delta_1 \cdots \delta_N \in \Delta^*$, and **non-erasing** if $\kappa(\delta) = \varepsilon \implies \delta = \varepsilon$. If $\kappa$ is multiplicative, it is prefix monotone; if it is also non-erasing, it is strict-prefix monotone. Both BPE and WordPiece are multiplicative and non-erasing.

## 3. Algorithms

This section gives algorithms for computing $p_\Sigma(\boldsymbol{\sigma})$, $\overrightarrow{p_\Sigma}(\boldsymbol{\sigma})$, $\overrightarrow{p_\Sigma}(\boldsymbol{\sigma}' \mid \boldsymbol{\sigma})$, $\overrightarrow{p_\Sigma}(\text{EOS} \mid \boldsymbol{\sigma})$, and conditional token generation. We assume throughout that $\kappa$ is strict-prefix monotone.

### 3.1. Covering

Eq. (11) shows that we can, in principle, compute the prefix probability $\overrightarrow{p_\Sigma}(\boldsymbol{\sigma})$ by summing over **prefix-encodings** of $\boldsymbol{\sigma}$, $\mathcal{P}(\boldsymbol{\sigma}) \stackrel{\text{def}}{=} \{\boldsymbol{\delta} \in \Delta^* : \kappa(\boldsymbol{\delta}) \succeq \boldsymbol{\sigma}\}$. Although $\mathcal{P}(\boldsymbol{\sigma})$ is infinitely large, we can exploit the prefix monotone structure of $\kappa$ to find a different way to perform the summation by summing over a *finite* set. Let $\boldsymbol{\delta} \in \Delta^*$, $\delta_1 \cdots \delta_M = \boldsymbol{\delta}$, and $\boldsymbol{\sigma} \in \Sigma^*$. We say that $\boldsymbol{\delta}$ **covers** $\boldsymbol{\sigma}$ if and only if $\kappa(\boldsymbol{\delta}) \succeq \boldsymbol{\sigma}$. Monotonicity ensures that for all $\boldsymbol{\delta} \in \mathcal{P}(\boldsymbol{\sigma})$, we have that $\forall \boldsymbol{\delta}' \in \Delta^* : \kappa(\boldsymbol{\delta} \cdot \boldsymbol{\delta}') \succeq \boldsymbol{\sigma}$. In other words, any $\boldsymbol{\delta}$ that decodes to an extension of $\boldsymbol{\sigma}$ (i.e., $\kappa(\boldsymbol{\delta}) \succeq \boldsymbol{\sigma}$) will continue to do so if we append tokens to it. Thus, we may additionally qualify the relationship as $\boldsymbol{\delta}$ **minimally covers** $\boldsymbol{\sigma}$ if additionally $\kappa(\delta_1 \cdots \delta_{M-1}) \prec \boldsymbol{\sigma}$. With that in mind, we define $\phi_{\boldsymbol{\sigma}}(\boldsymbol{\delta})$ as the *shortest* prefix $\boldsymbol{\delta}' \preceq \boldsymbol{\delta}$ such that $\kappa(\boldsymbol{\delta}') \succeq \boldsymbol{\sigma}$, i.e., $\phi_{\boldsymbol{\sigma}}$ maps any $\boldsymbol{\delta}$ that covers $\boldsymbol{\sigma}$ to a (possibly equal) token string that minimally covers $\boldsymbol{\sigma}$. Next, we define the set of *minimal* prefix encodings of $\boldsymbol{\sigma}$, which we call the **covering** of $\boldsymbol{\sigma}$, $\mathcal{C}(\boldsymbol{\sigma}) \stackrel{\text{def}}{=} \{\phi_{\boldsymbol{\sigma}}(\boldsymbol{\delta}) \mid \boldsymbol{\delta} \in \mathcal{P}(\boldsymbol{\sigma})\}$. A more convenient expression for the covering $\mathcal{C}(\boldsymbol{\sigma})$ of a

string $\sigma \in \Sigma^*$ is equal to the following subset of $\Delta^*$:

$$\mathcal{C}(\sigma) = \begin{cases} \{\varepsilon\} & \text{if } \sigma = \varepsilon \\ \{\delta_1 \cdots \delta_M \in \Delta^+: & \text{otherwise} \quad (12) \\ \quad \kappa(\delta_1 \cdots \delta_{M-1}) \prec \sigma \preceq \kappa(\delta_1 \cdots \delta_M)\} \end{cases}$$

**Example 2.**

*Recall the covering of the string $\sigma$ = Hello,␣worl from the introduction. We have repeated it on the right and couched it in our terminology. Note that the complete covering $\mathcal{C}$(Hello,␣worl) contains 36,608 token strings; we only show the top strings according to their respective $\overrightarrow{p_\Delta}$. Below, we list several properties and observations about the structure of the covering:*

$$\begin{bmatrix} \text{Hello} & , & \text{␣world} \\ 15496 & 11 & 995 \end{bmatrix}$$

$$\begin{bmatrix} \text{Hello} & , & \text{␣worlds} \\ 15496 & 11 & 11621 \end{bmatrix}$$

$$\begin{bmatrix} \text{Hello} & , & \text{␣worldwide} \\ 15496 & 11 & 8688 \end{bmatrix}$$

$$\begin{bmatrix} \text{Hello} & , & \text{␣worldly} \\ 15496 & 11 & 43249 \end{bmatrix}$$

$$\begin{bmatrix} \text{Hello} & , & \text{␣worldview} \\ 15496 & 11 & 29081 \end{bmatrix}$$

$$\begin{bmatrix} \text{Hell} & \text{o} & , & \text{␣world} \\ 28254 & 78 & 11 & 995 \end{bmatrix}$$

$$\begin{bmatrix} \text{Hello} & , & \text{␣wor} & \text{l} \\ 15496 & 11 & 476 & 75 \end{bmatrix}$$

- *The covering for any given string $\sigma$ always has the property that each non-empty token string $\delta$ in the covering decodes to a string that has $\sigma$ as a prefix, i.e., $\sigma \preceq \kappa(\delta)$. This is illustrated by the* gloss *string in the tokenization.*
- *Note that $\delta$ may include a* partially *matched token at its end (i.e., $\delta_M$ in Eq. (12)). We have marked the extra characters by* underlining *them. We note that the 7th member does not have a partially matched last token.*
- *Each token string in the covering has at most one partially matched token thanks to the condition $\kappa(\delta) \prec \sigma \preceq \kappa(\delta \cdot \delta)$. The 7th member of the cover has a completely matched last token; hence, there is no underlining.*
- *We see that if we were to extend any member $\delta \in \mathcal{C}(\sigma)$ with an arbitrary string of additional tokens $\delta'$, it would continue to decode to a string such that $\kappa(\delta \cdot \delta') \succeq \sigma$. Moreover, $\delta$ is minimal (i.e., $\phi_\sigma(\delta) = \delta$).*

The notion of a covering is used to derive an algorithm for computing character-level probabilities given a token-level language model. We first show how it gives us the prefix probability and subsequently give equations for the remaining quantities of the character-level language model.

**Proposition 1.** *Suppose $(\Sigma, \Delta, \tau, \kappa)$ is a tokenization model where $\kappa$ is strict-prefix monotone and $p_\Delta$ is a token-level language model. Then, the prefix probability $\overrightarrow{p_\Sigma}(\sigma)$ for the character-level model Eq. (6) is given by*

$$\overrightarrow{p_\Sigma}(\sigma) = \sum_{\delta \in \mathcal{C}(\sigma)} \overrightarrow{p_\Delta}(\delta), \qquad \forall \sigma \in \Sigma^* \quad (13)$$

*Proof.* See App. B. ∎

Eq. (13) is a substantial improvement over Eq. (7) for computing $\overrightarrow{p_\Sigma}(\sigma)$. Specifically, we now have a *finite* sum, as

$|\mathcal{C}(\sigma)|$ is finite for all $\sigma \in \Sigma^*$. Bear in mind that the covering's size is likely too large to be practical, as there may still be a large number of summands; however, the set of high-prefix-probability elements of the covering tends to be reasonably small, an observation that we verify in §4, and leverage to develop practical algorithms in §3.

Lastly, we note that the covering contains the set of encodings, i.e., $\mathcal{E}(\sigma) \subseteq \mathcal{C}(\sigma)$; hence, the encodings may be extracted from the covering as follows: $\mathcal{E}(\sigma) = \{\delta \in \mathcal{C}(\sigma): \kappa(\delta) = \sigma\}$. We may also express the probability of $\sigma$ in terms of the covering:

$$p_\Sigma(\sigma) = \sum_{\delta \in \mathcal{C}(\sigma)} \mathbb{1}\{\kappa(\delta) = \sigma\} p_\Delta(\delta) \quad (14)$$

**3.2. Algorithms for $\overrightarrow{p_\Sigma}(\sigma)$ and $p_\Sigma(\sigma)$**

The enumeration algorithm will enumerate elements of the covering along with their prefix probability (for convenience). It filters prefixes of token strings that cannot eventually cover the target string $\sigma$. The strict-prefix monotonicity property is essential for this filtering.

Our algorithm enum_cover performs recursive enumeration of the members of the covering $\mathcal{C}(\sigma)$ along with some metadata. Specifically, the algorithm returns a collection of triples where each triple $(p', \sigma', \delta')$ satisfies $\delta' \in \mathcal{C}(\sigma)$, $p' = \overrightarrow{p_\Delta}(\delta')$, and $\sigma' = \kappa(\delta')$.

```
5  def enum_cover(σ₁⋯σ_N):
6    if N = 0: return [(1, ε, ε)] # base case
7    out ← []
8    for (p', σ', δ') in enum_cover(σ₁⋯σ_{N-1}):
9      if |σ'| < N:  # extend
10       for δ'' ∈ Δ:
11         σ'' ← κ(δ'·δ'')
12         if σ''_N = σ_N: # filter
13           out.append((p' · p_Δ→(δ'' | δ'), σ'', δ'·δ''))
14     elif σ'_N = σ_N: # filter character matches
15       out.append((p', σ', δ'))
16   return prune(σ₁⋯σ_N, out)
```

Note that this method has an additional parameter, the function prune, which is used on the last line. This method, as the name suggests, is used to limit the size of the covering to prevent excessive growth. We will discuss this parameter shortly. For now, consider the following definition:

```
17  def prune_nothing(σ₁⋯σ_N, out):
18    return out
```

From the output of the enumeration algorithm, we can compute the other key objects and quantities (i.e., $\mathcal{C}(\sigma), \overrightarrow{p_\Sigma}(\sigma), \mathcal{E}(\sigma), p_\Sigma(\sigma)$) in the character-level interface.

**Time and space complexity.** To meaningfully discuss its running time, we assume the following:

- $\kappa(\boldsymbol{\delta}' \cdot \boldsymbol{\delta}'')$ can be evaluated in constant time given $\kappa(\boldsymbol{\delta}')$.
- the cost of evaluating $\overrightarrow{p_\Delta}(\delta_t \mid \boldsymbol{\delta}_{<t})$ is constant given that $\overrightarrow{p_\Delta}(\delta_s \mid \boldsymbol{\delta}_{<s})$ has been computed for $0 \leq s < t$.[13]

Under these assumptions, the running time of $\mathsf{enum\_cover}(\sigma_1 \cdots \sigma_N)$ can be exponential in $N$ when no pruning is used. We provide detailed bounds on the covering's size in App. C. It is straightforward to verify that the space complexity is $\mathcal{O}(|\mathcal{C}(\boldsymbol{\sigma})|)$, and the running time is $\mathcal{O}(|\Delta| \cdot \sum_{t=1}^{|\boldsymbol{\sigma}|} |\mathcal{C}(\boldsymbol{\sigma}_{<t})|)$ which is dominated by the $|\Delta| \cdot |\mathcal{C}(\boldsymbol{\sigma})|$ term; thus, $\mathcal{O}(|\Delta| \cdot |\mathcal{C}(\boldsymbol{\sigma})|)$.

**Pruning.** We now consider some useful pruning heuristics for the algorithm, which make it an *approximation* but substantially improve its running time. We propose a heuristic based on beam search. This heuristic is very effective: it gives us a linear running time as a function of the character string's length. It has a parameter $K$ that controls the approximation quality. Larger $K$ makes the approximation more accurate, and the approximation becomes exact as $K$ approaches the size of the (largest intermediate) covering. We take $K$ to be a global variable in the pseudocode.

$\hookrightarrow$ *Our pruning heuristic:* Our pruning heuristic enumerates $\leq K$ distinct token strings modulo their last token. This choice allows up to $|\Delta|$ versions of the last token to be enumerated (if it is not completely matched). Thus, the work done at each step is $\mathcal{O}(K \cdot |\Delta|)$, and the size of the pruned list is at most that size. Therefore, the overall running time is $\mathcal{O}(N \cdot K \cdot |\Delta|)$ for a character string of length $N$.[14]

```
19 def prune_top_K_buckets(σ₁⋯σN, results):
20     buckets ← {}
21     for item in results:
22         (p, σ′, δ₁⋯δM) ← item
23         # Exclude a partially matched last
24         key ← δ₁⋯δM₋₁ if |σ′| > N else δ₁⋯δM
25         buckets[key].append(item)
26     pruned ← []
27     for bucket in buckets.top(K): # by prob.
28         for item in bucket:
29             pruned.append(item)
30     return pruned
```

**Bundled beam summing implementation.** App. D describes an implementation strategy that improves the constant factors associated with the pseudocode above. The key idea is to group the token sequences that fall into the same bucket in $\mathsf{prune\_top\_K\_buckets}$ into a *bundle* that represents them compactly. In particular, we can use a trie

to efficiently filter out the next tokens that disagree with the next character. We can regard the trie as a local language model that generates the next token character-by-character according to the probability assigned by $\overrightarrow{p_\Delta}(\cdot \mid \boldsymbol{\delta})$. Each bundle can be unbundled (if necessary) into the respective tuples that the $\mathsf{enum\_cover}$ algorithm maintains.

### 3.3. Algorithms for $\overrightarrow{p_\Sigma}(\cdot \mid \boldsymbol{\sigma})$

This section gives algorithms for computing the character-level conditional prefix probability. Recall the definition of the character-level conditional prefix probability, that is, Eq. (8) and (9), can be computed from a certain ratio of calls to $\overrightarrow{p_\Sigma}$ (and $p_\Sigma$ in the case of EOS). From here, Eq. (8) and (9) give a straightforward algorithm for computing the distribution over $\Sigma \cup \{\text{EOS}\}$ given $\boldsymbol{\sigma} \in \Sigma^*$. However, a direct translation would perform duplicate work. Therefore, we provide the following version, which reuses work between the calls to $\overrightarrow{p_\Sigma}$ than directly evaluating those equations would do.

```
31 def next_character_probability(σ):
32     N ← |σ|;  Z ← 0;  p̄ ← {σ′: 0 for σ′ ∈ Σ∪{EOS}}
33     for (p′, σ′, δ) ∈ enum_cover(σ):
34         Z += p′
35         if |σ′| = N:  # i.e., σ′ = σ
36             p̄(EOS) += p′ · p→_Δ(EOS | δ)
37             for δ″ ∈ Δ:  # extend
38                 σ″ ← κ(δ′·δ″)
39                 p̄(σ″_{N+1}) += p′ · p→_Δ(δ″ | δ′)
40         else:  # i.e., σ′ ⪰ σ
41             p̄(σ′_{N+1}) += p′
42     return p̄/Z      # Z = p→_Σ(σ)
```

### 3.4. Conditional Generation $p_{\Delta|\Sigma}(\boldsymbol{\delta} \mid \boldsymbol{\sigma})$

This section gives a simple algorithm for correctly generating a token string $Y$ that has a given character-level prompt $\boldsymbol{\sigma}$ as its prefix. This algorithm is equivalent to the algorithm in the introduction but significantly faster.

The algorithm works by enumerating the covering $\mathcal{C}(\boldsymbol{\sigma})$, drawing a token string from it in proportion to its prefix probability, and finishing the token string by sampling a completion, which can be done from the token-level model.

```
43 def conditional_token_generation(σ):
44     δ′ ~ Categorical({δ′: p′/p→_Σ(σ)
45             for (p′, _, δ′) in enum_cover(σ)})
46     return sample_completion(δ′)

47 def sample_completion(δ′):
48     δ″ ← ε
49     while True:
50         δ ~ p→_Δ(· | δ′·δ″)
51         if δ = EOS: break
52         δ″ ← δ″·δ
53     return δ′·δ″
```

---

[13]In the case of the common transformer language model (Vaswani et al., 2017), this can be achieved with efficient caching and limiting context windows to a constant size.

[14]Note: finding the (unordered) set of top-$K$ elements from a set $S$ is possible in $\mathcal{O}(|S|)$ time via the median-of-medians algorithm.

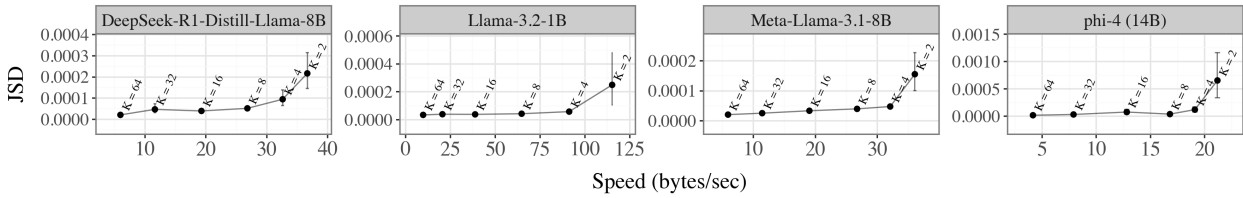

(a) Error (JSD/byte) vs. speed (bytes/sec). Error is computed between the character-level conditional distributions with beam sizes $K \in \{2, 4, 8, 16, 32, 64\}$ and a reference distribution computed using a much larger value of $K = 128$.

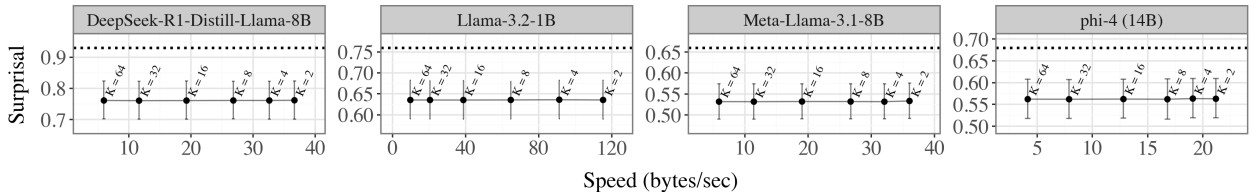

(b) Surprisal (bits/byte) vs. speed (bytes/sec). The dotted line shows the canonically tokenized baseline's surprisal.

Figure 1: Experimental results. Averages are computed over the first 4000 bytes of the `wikitext-103-v1` test set. Error bars denote bootstrapped 95% confidence intervals. See App. A for the full table of results.

The following proposition establishes correctness:

**Proposition 2.** `conditional_token_generation`$(\sigma)$ *generates samples according to* $p_{\Delta|\Sigma}(\cdot \mid \sigma)$ *for all* $\sigma \in \Sigma^*$.

*Proof.* See App. E. ∎

We also note the following corollary, as it gives an interpretation for the categorical distribution in the efficient `conditional_token_generation` algorithm.

**Corollary 1.** *For all* $\sigma \in \Sigma^*$, $\delta \in \Delta^*$,

$$\mathbb{P}_{Y \sim p_\Delta}[\phi_\sigma(Y) = \delta \mid \kappa(Y) \succeq \sigma] = \frac{\overrightarrow{p_\Delta}(\delta)}{\overrightarrow{p_\Sigma}(\sigma)} \mathbb{1}\{\delta \in \mathcal{C}(\sigma)\} \quad (15)$$

Thus, we have provided an efficient solution to the prompt boundary problem. We also note that generating from $p_\Sigma$ a character at a time is also a correct solution to the prompt boundary problem; however, it is slower, as it does not benefit from the fact that the generated string is shorter in token space. This is because once the minimally covering token string has been sampled, the method `sample_completion` will generate a complete sequence more efficiently than the character-at-a-time sample algorithm.

## 4. Experiments

This section investigates our algorithm's running time and accuracy. We use the following setup:

- We use the following publicly available models: Llama-3.2-1B, Meta-Llama-3.1-8B, DeepSeek-R1-Distill-Llama-8B, and phi-4 (14B) from the 🤗 transformers library (Wolf et al., 2020). Each model was trained over

token strings created from byte-pair encoding (BPE; Sennrich et al. (2016); Gage (1994)).[15]

- We use the `wikitext-103-v1` corpus as a source of character strings; we used the version in the 🤗 datasets library. Specifically, we use the *test* portion.
- We use the **GenLM** library[16] with the v**LLM** (Kwon et al., 2023) backend to perform the efficient, batched evaluation of transformer language models on GPUs. We batch-evaluate all sequence extensions. Experiments were run on an L40S GPU with 40GB of memory.
- Our implementation utilizes a trie to efficiently represent all items in each bucket (see App. D), and the bucket-based pruning heuristic (§3).

To better understand the quality of the approximation our method provides, we perform the following experiments:[17]

- We measure the approximation error as the average **Jensen–Shannon distance** (**JSD**) to a reference model's conditional distribution over the next byte (Fig. 1a). We use a large beam $K = 128$ as a reference model.
- We evaluate the average **surprisal** ($-\log_2$ probability) of our model's estimated conditional distribution over the next byte in the corpus. As a baseline, we use the average surprisal (bits/bytes) of the canonical tokenization under the token-level language model (Fig. 1b).

---

[15] Note that in this section we use *bytes* as our set of characters to be compatible with the byte-pair encoding algorithm.

[16] `https://github.com/genlm/genlm-backend`

[17] Some additional baselines and experimental results are included in appendices (F and G).

**Discussion.** As expected, we observe that the speed (bytes/sec) decreases as $K$ increases. We observe an inverse relationship between error (JSD/byte) and speed (bytes/sec): as the processing speed (bytes/sec) decreases, the error also decreases. Notably, this tradeoff is non-linear, with error increasing more sharply at higher processing speeds compared to lower speeds. This trend is evident in all models; in general, error appears to flatten out after $K \geq 8$. This indicates diminishing returns in reducing error as $K$ gets large. We hypothesize that this occurs because the language model's probability mass is concentrated on a limited set of tokenizations, which are adequately covered even with smaller beam sizes. We also observe (as expected) that larger models run more slowly; however, this appears to be due to their higher evaluation time rather than the need for larger covers.

We also observe that the average per-byte surprisal is significantly lower under all models than the canonically tokenized baseline. The reasons for this are twofold: (1) Each model assigns non-negligible probability to noncanonical tokenizations of the corpus, which are being thrown out in the baseline estimate, but that is accounted for in our estimate. (2) The most likely tokenization of the corpus is often noncanonical, which our method is better able to find, as our beam-summing method uses the probabilities assigned to tokenizations. In contrast, the baseline uses only the hard-coded canonical tokenization. Interestingly, increasing $K$ does not appear to significantly decrease the surprisal, which we suspect is because the relatively greedy ($K = 2$) tokenizations adequately cover it.[18]

## Conclusion

We have developed an effective method for ameliorating tensions between tokens and characters faced by engineers and users. We gave theory and algorithms that provide a character-level interface to tokenized language models. We characterized and resolved the prompt boundary problem. We investigated the empirical speed and error rates of our method on two modern language models. The primary limitation of our beam summing method is that it requires a very large beam size $K$ if the language model does not favor a small number of tokenizations. The models that we explored in our experiments concentrate mass on a few tokenizations; thus, we did not require large $K$ to estimate their character-level prefix probabilities accurately.

## Acknowledgments

The authors would like to thank Andreas Opedal, Alex Lew, Ben Lipkin, Jacob Hoover Vigly, Luca Malagutti, Manuel de Prada Corral, Vésteinn Snæbjarnarson, Samuel Kiegeland, and Yahya Emara for their helpful feedback and discussions. JT would like to thank Rycolab for its hospitality during a recent visit. The authors JLG and JT would like to thank Institut des Hautes Études Scientifiques (IHES) for their hospitality while revising this paper. MG was supported by an ETH Zürich Postdoctoral Fellowship. This research was enabled in part by compute resources provided by Mila (mila.quebec).

## Impact Statement

The primary impact of this work is a more predictable, user-friendly interface for working with tokenized language models. By ensuring tools behave as expected, we mitigate certain unintended behaviors, such as the prompt-boundary problem highlighted in the introduction. More broadly, we do not anticipate any additional risks beyond those already inherent in the use of tokenized language models.

## Limitations

Our method should work well with other tokenized language models, provided they were trained with a *deterministic* tokenizer. However, if they were trained with a *stochastic* tokenizer, such as UnigramLM (Kudo, 2018) or BPE-Dropout (Provilkov et al., 2020), we would expect that the probability mass over tokenizations in each covering would not be heavily concentrated on a small subset. Thus, these models may require a large beam size, making our approach expensive. Sampling-based methods may provide a better speed–accuracy tradeoff for this setting.

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

# A. Supplementary Results

This section presents the data for Fig. 1 in tabular form.

## A.1. Jensen–Shannon Distance

| $K$ | Llama-3.2-1B | | Meta-Llama-3.1-8B | |
|---|---|---|---|---|
| | JSD / byte | bytes / sec | JSD / byte | bytes / sec |
| 2 | 0.00025 (0.00010, 0.00048) | 115.18 (110.54, 117.94) | 0.00016 (0.00010, 0.00023) | 36.03 (35.90, 36.16) |
| 4 | 0.00006 (0.00005, 0.00007) | 91.22 (88.36, 92.97) | 0.00005 (0.00004, 0.00005) | 32.06 (31.74, 32.27) |
| 8 | 0.00004 (0.00004, 0.00005) | 64.71 (63.50, 65.55) | 0.00004 (0.00004, 0.00004) | 26.72 (26.43, 26.95) |
| 16 | 0.00004 (0.00003, 0.00004) | 38.77 (37.77, 39.62) | 0.00003 (0.00003, 0.00004) | 19.03 (18.76, 19.28) |
| 32 | 0.00004 (0.00004, 0.00004) | 20.53 (19.93, 21.11) | 0.00003 (0.00002, 0.00003) | 11.42 (11.21, 11.62) |
| 64 | 0.00003 (0.00003, 0.00004) | 9.77 (9.52, 10.02) | 0.00002 (0.00002, 0.00002) | 5.91 (5.81, 6.02) |
| 128 | *(not applicable)* | 4.91 (4.81, 5.02) | *(not applicable)* | 2.97 (2.93, 3.02) |

| $K$ | DeepSeek-R1-Distill-Llama-8B | | phi-4 (14B) | |
|---|---|---|---|---|
| | JSD / byte | bytes / sec | JSD / byte | bytes / sec |
| 2 | 0.00022 (0.00015, 0.00032) | 36.64 (36.47, 36.83) | 0.00065 (0.00034, 0.00116) | 21.17 (21.03, 21.30) |
| 4 | 0.00009 (0.00007, 0.00014) | 32.60 (32.24, 32.87) | 0.00012 (0.00008, 0.00017) | 19.09 (18.98, 19.18) |
| 8 | 0.00005 (0.00004, 0.00006) | 26.81 (26.52, 27.04) | 0.00004 (0.00003, 0.00006) | 16.79 (16.69, 16.88) |
| 16 | 0.00004 (0.00003, 0.00005) | 19.26 (18.99, 19.52) | 0.00008 (0.00004, 0.00012) | 12.81 (12.68, 12.94) |
| 32 | 0.00005 (0.00003, 0.00007) | 11.62 (11.42, 11.83) | 0.00003 (0.00003, 0.00004) | 7.90 (7.78, 8.01) |
| 64 | 0.00002 (0.00002, 0.00002) | 5.96 (5.86, 6.07) | 0.00002 (0.00002, 0.00002) | 4.16 (4.10, 4.22) |
| 128 | *(not applicable)* | 3.01 (2.97, 3.06) | *(not applicable)* | 2.44 (2.40, 2.48) |

## A.2. Surprisal

| Model | Canonically Tokenized Baseline bits / byte |
|---|---|
| Llama-3.2-1B | 0.76 (0.69, 0.83) |
| Meta-Llama-3.1-8B | 0.66 (0.59, 0.74) |
| DeepSeek-R1-Distill-Llama-8B | 0.93 (0.83, 1.03) |
| phi-4 (14B) | 0.68 (0.61, 0.76) |

| K | Llama-3.2-1B bits / byte | bytes / sec | Meta-Llama-3.1-8B bits / byte | bytes / sec |
|---|---|---|---|---|
| 2 | 0.63539 (0.58988, 0.68280) | 115.18 (110.54, 117.94) | 0.53339 (0.49164, 0.57577) | 36.03 (35.90, 36.16) |
| 4 | 0.63593 (0.59076, 0.68238) | 91.22 (88.36, 92.97) | 0.53173 (0.49087, 0.57468) | 32.06 (31.74, 32.27) |
| 8 | 0.63540 (0.58932, 0.68020) | 64.71 (63.50, 65.55) | 0.53202 (0.49109, 0.57463) | 26.72 (26.43, 26.95) |
| 16 | 0.63518 (0.58985, 0.68186) | 38.77 (37.77, 39.62) | 0.53208 (0.49058, 0.57452) | 19.03 (18.76, 19.28) |
| 32 | 0.63524 (0.58988, 0.68187) | 20.53 (19.93, 21.11) | 0.53176 (0.49012, 0.57411) | 11.42 (11.21, 11.62) |
| 64 | 0.63525 (0.59041, 0.68259) | 9.77 (9.52, 10.02) | 0.53180 (0.48991, 0.57464) | 5.91 (5.81, 6.02) |
| 128 | 0.63550 (0.59101, 0.68157) | 4.91 (4.81, 5.02) | 0.53175 (0.49091, 0.57396) | 2.97 (2.93, 3.02) |

| K | DeepSeek-R1-Distill-Llama-8B bits / byte | bytes / sec | phi-4 (14B) bits / byte | bytes / sec |
|---|---|---|---|---|
| 2 | 0.76195 (0.70150, 0.82342) | 36.64 (36.47, 36.83) | 0.56296 (0.51968, 0.60938) | 21.17 (21.03, 21.30) |
| 4 | 0.76134 (0.70160, 0.82244) | 32.60 (32.24, 32.87) | 0.56345 (0.51961, 0.60944) | 19.09 (18.98, 19.18) |
| 8 | 0.76136 (0.70270, 0.82294) | 26.81 (26.52, 27.04) | 0.56194 (0.51601, 0.60854) | 16.79 (16.69, 16.88) |
| 16 | 0.76101 (0.70137, 0.82419) | 19.26 (18.99, 19.52) | 0.56228 (0.51854, 0.60802) | 12.81 (12.68, 12.94) |
| 32 | 0.76080 (0.70129, 0.82283) | 11.62 (11.42, 11.83) | 0.56192 (0.51807, 0.60742) | 7.90 (7.78, 8.01) |
| 64 | 0.76128 (0.70313, 0.82384) | 5.96 (5.86, 6.07) | 0.56206 (0.51752, 0.60760) | 4.16 (4.10, 4.22) |
| 128 | 0.76093 (0.70029, 0.82178) | 3.01 (2.97, 3.06) | 0.56185 (0.51753, 0.60815) | 2.44 (2.40, 2.48) |

## B. Proof of Proposition 1

**Proposition 1.** *Suppose $(\Sigma, \Delta, \tau, \kappa)$ is a tokenization model where $\kappa$ is strict-prefix monotone and $p_\Delta$ is a token-level language model. Then, the prefix probability $\overrightarrow{p_\Sigma}(\boldsymbol{\sigma})$ for the character-level model Eq. (6) is given by*

$$\overrightarrow{p_\Sigma}(\boldsymbol{\sigma}) = \sum_{\boldsymbol{\delta} \in \mathcal{C}(\boldsymbol{\sigma})} \overrightarrow{p_\Delta}(\boldsymbol{\delta}), \qquad \forall \boldsymbol{\sigma} \in \Sigma^* \tag{13}$$

*Proof.* We prove the proposition through the following manipulations.

$$\overrightarrow{p_\Sigma}(\boldsymbol{\sigma}) = \sum_{\boldsymbol{\delta}' \in \Delta^*} \mathbb{1}\{\boldsymbol{\sigma} \preceq \kappa(\boldsymbol{\delta}')\} p_\Delta(\boldsymbol{\delta}') \tag{16}$$

$$= \mathbb{1}\{\boldsymbol{\sigma} = \varepsilon\} p_\Delta(\varepsilon) + \sum_{\boldsymbol{\delta}' \in \Delta^+} \mathbb{1}\{\boldsymbol{\sigma} \preceq \kappa(\boldsymbol{\delta}')\} p_\Delta(\boldsymbol{\delta}') \tag{17}$$

$$= \mathbb{1}\{\boldsymbol{\sigma} = \varepsilon\} p_\Delta(\varepsilon) + \sum_{\boldsymbol{\delta} \cdot \boldsymbol{\delta}' \cdot \boldsymbol{\delta}'' \in \Delta^+} \mathbb{1}\{\kappa(\boldsymbol{\delta}) \prec \boldsymbol{\sigma} \preceq \kappa(\boldsymbol{\delta} \cdot \boldsymbol{\delta}' \cdot \boldsymbol{\delta}'')\} p_\Delta(\boldsymbol{\delta} \cdot \boldsymbol{\delta}' \cdot \boldsymbol{\delta}'') \tag{18}$$

$$= \mathbb{1}\{\boldsymbol{\sigma} = \varepsilon\} p_\Delta(\varepsilon) + \sum_{\boldsymbol{\delta} \cdot \boldsymbol{\delta}' \in \Delta^+} \mathbb{1}\{\kappa(\boldsymbol{\delta}) \prec \boldsymbol{\sigma} \preceq \kappa(\boldsymbol{\delta} \cdot \boldsymbol{\delta}')\} \sum_{\boldsymbol{\delta}'' \in \Delta^*} p_\Delta(\boldsymbol{\delta} \cdot \boldsymbol{\delta}' \cdot \boldsymbol{\delta}'') \tag{19}$$

$$= \mathbb{1}\{\boldsymbol{\sigma} = \varepsilon\} p_\Delta(\varepsilon) + \sum_{\boldsymbol{\delta} \cdot \boldsymbol{\delta}' \in \Delta^+} \mathbb{1}\{\kappa(\boldsymbol{\delta}) \prec \boldsymbol{\sigma} \preceq \kappa(\boldsymbol{\delta} \cdot \boldsymbol{\delta}')\} \overrightarrow{p_\Delta}(\boldsymbol{\delta} \cdot \boldsymbol{\delta}') \tag{20}$$

$$= \sum_{\boldsymbol{\delta} \in \mathcal{C}(\boldsymbol{\sigma})} \overrightarrow{p_\Delta}(\boldsymbol{\delta}) \tag{21}$$

About the steps above: We start with the summation expression for the character-level prefix probability (i.e., Eq. (11)). We expand the summation into two cases (so that it will eventually match the two cases in the expression for the covering Eq. (12)). Next, for each summand, we consider its *unique* minimal prefix $\boldsymbol{\delta} \cdot \boldsymbol{\delta}'$ covering $\boldsymbol{\sigma}$. We see why $\varepsilon$ is handled separately, as it cannot be covered by a token sequence of that form. We exploit the key property of prefix monotone tokenizers (i.e., that once $\boldsymbol{\delta} \cdot \boldsymbol{\delta}'$ covers $\boldsymbol{\sigma}$, each extension $\boldsymbol{\delta} \cdot \boldsymbol{\delta}' \boldsymbol{\delta}''$ continues to cover it). This allows us to rearrange the summation to sum over the extension $\boldsymbol{\delta}''$, which is conveniently equal to the prefix probability of $\boldsymbol{\delta} \cdot \boldsymbol{\delta}'$. The final step is to recognize that the summands can all be indexed by the covering $\mathcal{C}(\boldsymbol{\sigma})$. ∎

## C. The Size of the Covering

We now set about to bound the worst-case size of the covering function. To do so, we introduce additional definitions that characterize the different growth factors.

We define $\kappa$'s **fertility** as

$$F \stackrel{\text{def}}{=} \max_{\boldsymbol{\delta} \in \Delta^*} \max_{\boldsymbol{\sigma} \in \Sigma^*} |\{\delta' \in \Delta : \boldsymbol{\sigma} = \kappa(\boldsymbol{\delta} \cdot \delta')\}| \leq |\Delta| \tag{22}$$

**Example 3.** *The BPE tokenizer has $F_{\text{BPE}} = 1$ because it is multiplicative, and its tokens represent* distinct *substrings. More formally,*

$$\begin{aligned} F_{\text{BPE}} &= \max_{\boldsymbol{\delta} \in \Delta^*} \max_{\boldsymbol{\sigma} \in \Sigma^*} |\{\delta' \in \Delta : \boldsymbol{\sigma} = \kappa(\boldsymbol{\delta} \cdot \delta')\}| & \textit{[def of fertility]} \tag{23} \\ &= \max_{\boldsymbol{\delta} \in \Delta^*} \max_{\boldsymbol{\sigma} \in \Sigma^*} |\{\delta' \in \Delta : \boldsymbol{\sigma} = \kappa(\boldsymbol{\delta}) \cdot \kappa(\delta')\}| & \textit{[def multiplicativity]} \tag{24} \\ &= |\{\delta' \in \Delta : \boldsymbol{\sigma}' = \kappa(\delta')\}| & \textit{[def function]} \tag{25} \\ &= 1 & \textit{[distinctness]} \tag{26} \end{aligned}$$

Additionally, we define a $\kappa$'s **munch** as follows.

$$M \stackrel{\text{def}}{=} \max_{\boldsymbol{\delta} \in \Delta^*} \max_{\delta \in \Delta} |\kappa(\boldsymbol{\delta} \cdot \delta)| - |\kappa(\boldsymbol{\delta})| \tag{27}$$

In words, the munch measures the length of the largest number of characters that can be introduced by adding one more token to any given context.

**Example 4.** *The munch of a multiplicative $\kappa$, such as BPE, is $\max_{\delta \in \Delta} |\kappa(\delta)|$. Put in words, it is the length of the longest detokenization. The munch for* GPT-2 *is surprisingly long (128).*

**Proposition 3.** *Let $F$ and $M$ be the fertility and munch of $\kappa$. Then, for all $\boldsymbol{\sigma} \in \Sigma^*$,*

$$|\mathcal{C}(\boldsymbol{\sigma})| \leq C(|\boldsymbol{\sigma}|) \tag{28}$$

*where*

$$C(n) \stackrel{\text{def}}{=} \begin{cases} 0 & \textit{if } n < 0 \\ 1 & \textit{if } n = 0 \\ F \sum_{j=n-M}^{n-1} C(j) & \textit{otherwise)} \end{cases} \tag{29}$$

*Proof.* The base cases $N \leq 0$ are straightforward. Consider the case of a string of length $N \geq 0$. *Inductive hypothesis*: Suppose for all strings $\boldsymbol{\sigma}'$ with $|\boldsymbol{\sigma}'| < N$, $|\mathcal{C}(\boldsymbol{\sigma}')| \leq C(|\boldsymbol{\sigma}'|)$.

Let $\boldsymbol{\sigma}$ be an arbitrary string with length $N > 0$.

$$|\mathcal{C}(\sigma_1 \cdots \sigma_N)| \tag{30}$$

$$= \left|\left\{\boldsymbol{\delta}\cdot\delta \in \Delta^+ : \kappa(\boldsymbol{\delta}) \prec \sigma_1 \cdots \sigma_N \preceq \kappa(\boldsymbol{\delta}\cdot\delta)\right\}\right| \tag{31}$$

$$= \left|\bigcup_{j=0}^{N} \underbrace{\left\{\boldsymbol{\delta}\cdot\delta \in \Delta^+ : \kappa(\boldsymbol{\delta}) = \sigma_1 \cdots \sigma_j, \sigma_1 \cdots \sigma_N \preceq \kappa(\boldsymbol{\delta}\cdot\delta)\right\}}_{=\emptyset \text{ if } N-(j+1)>M \text{ or } N=j+1}\right| \tag{32}$$

$$\leq \sum_{j=N-M}^{N-1} \underbrace{\left|\left\{\boldsymbol{\delta} \in \Delta^* : \kappa(\boldsymbol{\delta}) = \sigma_1 \cdots \sigma_j\right\}\right.}_{\subseteq \mathcal{C}(\sigma_1 \cdots \sigma_j)} \cdot \underbrace{\left|\left\{\delta \in \Delta : \boldsymbol{\delta} \in \Delta^*, \sigma_1 \cdots \sigma_N \preceq \kappa(\boldsymbol{\delta}\cdot\delta)\right\}\right|}_{\leq F} \tag{33}$$

$$\leq F \cdot \sum_{j=N-M}^{N} \underbrace{|\mathcal{C}(\sigma_1 \cdots \sigma_j)|}_{\text{inductive hypothesis}} \tag{34}$$

$$\leq F \cdot \sum_{j=N-M}^{N} C(j) \tag{35}$$

$$= C(N) \tag{36}$$

Thus, the proposition holds true by the principle of induction. ∎

**Corollary 2.** *Let $N = |\boldsymbol{\sigma}|$. Consider the following cases:*

- *When $M = N$ and $F = 1$, $C(N) = 2^N$.*
- *When $M = N$ and $F \geq 0$, $C(N) = F(1 + F)^N$.*
- *Otherwise, $C(N) = F^N \mathrm{Fib}(N, M)$ where $\mathrm{Fib}(N, M)$ is the $N^{th}$ $M^{th}$-order Fibonacci number.*[19]

*In all cases, $C_M^F(N) < \infty$.*

In the proposition below, we show that the covering can easily be exponential in size:

**Proposition 4.**

$$|\mathcal{C}(\boldsymbol{\sigma})| \in \Omega(2^{|\boldsymbol{\sigma}|}) \tag{37}$$

*Proof.* We prove the proposition by constructing an example that achieves the lower bound.

- Let $\Sigma = \{\mathsf{a}\}$, $\Delta = \left\{\underset{1}{\mathsf{a}}, \underset{2}{\mathsf{aa}}\right\}$.
- Let $\kappa$ be multiplicative, and define $\kappa\binom{\mathsf{a}}{1} \overset{\text{def}}{=} \mathsf{a}$, $\kappa\binom{\mathsf{aa}}{2} \overset{\text{def}}{=} \mathsf{aa}$.
- Let $\boldsymbol{\sigma}$ be an arbitrary string from $\Sigma^*$. Let $N = |\boldsymbol{\sigma}|$.

Then, $|\mathcal{E}(\boldsymbol{\sigma})| \geq$ the number of nonnegative integer solutions $(n, m)$ to $1\,m + 2\,n = N$. This is because we can build the $\mathsf{a}^N$ using a sequence of $\underset{1}{\mathsf{a}}$ or $\underset{2}{\mathsf{aa}}$, but each $\underset{1}{\mathsf{a}}$ accounts for 1 $\mathsf{a}$ and each $\underset{2}{\mathsf{aa}}$ accounts for 2. So if $n$ is the number of token 1 and $m$ is the number of token 2, we must have that $1\,m + 2\,n = N$. The number of solutions grows like $\Omega(2^N)$. Lastly, because $\mathcal{C}(\boldsymbol{\sigma}) \supseteq \mathcal{E}(\boldsymbol{\sigma})$, we have that $|\mathcal{C}(\boldsymbol{\sigma})| \in \Omega(2^{|\boldsymbol{\sigma}|})$. Thus, the proposition holds. ∎

---

[19] $M^{th}$-order Fibonacci numbers are a variation of the well-known Fibonacci (i.e., $M = 2$) that sums the previous $M$ numbers in the sequence instead of the previous two.

## D. Bundled Beam Summing Implementation

Upon implementing this scheme, we observed that it is possible to efficiently reason about all the next tokens that extend a given token sequence in the cover in bulk. The key idea is to group the token sequences that fall into the same bucket in `prune_top_K_buckets` into a `Bundle` (see below) that represents them compactly. In particular, we can use a probability trie to efficiently filter out the next tokens that disagree with the next character. This improves the per-iteration cost of that filter as the data structure does the organizational work ahead of time (in bulk). We can regard the probability trie as a local language model that generates the next token character-by-character according to the probability assigned to it by $\overrightarrow{p_\Delta}(\cdot \mid \delta)$. Each bundle can be unbundled (see method `unbundle`) into the respective tuples that the `enum_cover` algorithm maintains. The algorithms are otherwise equivalent.

```
54  def beam(σ₁ ··· σ_N):
55    if N = 0: return [Bundle(1, ε, ε, build_trie(ε))]
56    candidates ← []
57    for bundle in beam(σ₁ ··· σ_{N−1}):
58      filtered_bundle ← bundle.filter(σ_N)
59      if filtered_bundle is not None:
60        candidates.append(filtered_bundle)
61      for extended_bundle in bundle.extend():
62        candidates.append(extended_bundle.filter(σ_N))
63    # Keep top-K bundles according to their prefix probability
64    return top_K(candidates, key=lambda bundle: –bundle.p)
```

Each bundle is an instance of the following class with four member variables: the prefix probability $p$, token string $\delta$, character string $\sigma$, and a reference to a local probability trie $trie$. The trie provides the character-level probabilities of the distribution over possible next tokens: $p_\Delta(\cdot \mid \delta)$. The trie is also augmented with a special symbol EOT to denote the end of this next token.[20]

```
65  class Bundle(p, δ, σ, trie):
66
67    def filter(σ′):
68      if trie.p(σ′ | σ) = 0: return    # no tokens give prefix σ·σ′ prob
69      return Bundle(p · trie(σ′ | σ), δ, σ·σ′, trie)
70
71    def extend():
72      Z ← trie.p(EOT | σ)
73      if Z > 0:  # emit tokens that decode to σ
74        for (δ′, p′) in trie.tokens[σ]:
75          yield Bundle(p/Z · p′, δ·δ′, ε, build_trie(δ·δ′))
```

The code below builds the probability trie from the possible next tokens. It figures out the character strings associated with those next tokens and puts them into the trie with the respective probabilities.[21]

```
76  def build_trie(δ):
77    σ ← κ(δ)                     # remember common prefix
78    trie ← ProbabilityTrie()   # uses EOT to mark the end of a token
79    for δ′ ∈ Δ:
80      σ·σ′ ← κ(δ·δ′)             # use new characters, i.e., ignore prefix σ
81      trie.add(σ′, δ′, p_Δ(δ′ | δ))   # add σ′ and δ′ to trie with this probability
82    return trie
```

The probability trie has the following functionality:

- $trie.tokens[\sigma']$ stores the set of tokens that decode to $\sigma'$ along with their respective mass.

---

[20]This is completely analogous to how EOS marks the end of a string.

[21]Note that EOS is handled as an indivisible symbol in the probability trie, whereas other tokens can be expanded into characters (or bytes).

- $trie.p(\sigma' \mid \boldsymbol{\sigma})$ returns the probability of the character $\sigma'$ given $\boldsymbol{\sigma}$, it is proportional to

$$trie.p(\sigma' \mid \boldsymbol{\sigma}) \propto \sum_{\delta' \in \Delta} p_\Delta(\delta' \mid \boldsymbol{\delta}) \begin{cases} \mathbb{1}\{\kappa(\boldsymbol{\delta}\cdot\delta') = \boldsymbol{\sigma}\} & \textbf{if } \sigma' = \text{EOT} \\ \mathbb{1}\{\kappa(\boldsymbol{\delta}\cdot\delta') \succeq \boldsymbol{\sigma}\cdot\sigma'\} & \textbf{otherwise} \end{cases} \tag{38}$$

We note that the trie implicitly depends on the token string $\boldsymbol{\delta}$ used in its creation.

To aid in understanding how the bundled algorithm relates to the original algorithm, we give the following methods.

```
83  class Bundle(p, δ, σ, trie):
84    ...
85    def unbundle():
86      agenda ← [σ]
87      while agenda:
88        σ' ← agenda.pop()
89        Z ← trie.p(EOT | σ')
90        if Z > 0:
91          for (δ', p') ∈ trie.tokens[σ']:
92            yield (p · p', κ(δ·δ'), δ·δ')
93        for σ'' in trie.p(· | σ'):
94          if σ'' ≠ EOT:
95            agenda.append(σ'·σ'')

96  def unbundle_beam(beam):
97    return [item for bundle in beam for item in bundle.unbundle()]
```

We note that unbundle_beam(beam($\boldsymbol{\sigma}$)) gives precisely the same set of elements as the unbundled algorithm (i.e., enum_cover($\boldsymbol{\sigma}$)) run on the same string $\boldsymbol{\sigma}$ and the bucket-based pruning scheme with parameter $K$ (up to reordering).

To compute the next-character probability, we use the following algorithm:

```
98   def next_character_probability(σ, method=beam):
99     p̄ ← {σ': 0 for σ' ∈ Σ ∪ {EOS}}
100    for bundle in method(σ):  # use beam approximation by default
101      for σ' ∈ Σ:
102        p̄(σ') += bundle.p · bundle.trie.p(σ' | bundle.σ)
103      for ext_bundle in bundle.extend():
104        for σ' ∈ Σ:
105          p̄(σ') += ext_bundle.p · ext_bundle.trie.p(σ' | ext_bundle.σ)
106    Z ← sum(p̄.values())
107    return {σ': p̄(σ')/Z for σ' ∈ Σ}
```

# E. Proof of Proposition 2

**Proposition 2.** `conditional_token_generation(`$\sigma$`)` *generates samples according to* $p_{\Delta|\Sigma}(\cdot \mid \sigma)$ *for all* $\sigma \in \Sigma^*$.

*Proof (Sketch).* Choose an arbitrary $\delta \in \Delta^*$.

$$p_{\Delta|\Sigma}(\delta \mid \sigma) = \mathbb{P}_{Y \sim p_\Delta}[Y = \delta \mid \kappa(Y) \succeq \sigma] \tag{39}$$

$$= p_\Delta(\delta) \frac{\mathbb{1}\{\kappa(\delta) \succeq \sigma\}}{\mathbb{P}_{Y \sim p_\Delta}[\kappa(Y) \succeq \sigma]} \tag{40}$$

$$= p_\Delta(\delta) \frac{\mathbb{1}\{\kappa(\delta) \succeq \sigma\}}{\overrightarrow{p_\Sigma}(\sigma)} \tag{41}$$

Let $\delta' = \phi_\sigma(\delta)$ (i.e., the shortest prefix of $\delta$ such that $\kappa(\delta') \succeq \sigma$). Choose $\delta''$ such that $\delta' \cdot \delta'' = \delta$.

$$= \underbrace{\overrightarrow{p_\Delta}(\text{EOS} \mid \delta' \cdot \delta'') \overrightarrow{p_\Delta}(\delta'' \mid \delta')}_{\text{sample completion}} \underbrace{\overrightarrow{p_\Delta}(\delta') \frac{\mathbb{1}\{\kappa(\delta') \succeq \sigma\}}{\overrightarrow{p_\Sigma}(\sigma)}}_{\text{sample from covering}} \tag{42}$$

We can see that the algorithm samples from this distribution because it samples a token string $\delta'$ from the covering in proportion to the right factor in Eq. (42) and then samples a completion $\delta''$ of $\delta'$ in proportion to the left factor of the equation. Thus, the sample $\delta = \delta' \cdot \delta''$ has probability $p_{\Delta|\Sigma}(\delta \mid \sigma)$, and `conditional_token_generation(`$\sigma$`)` is a correct sampling procedure for it. ∎

## F. Token Healing Baseline

We implemented an additional baseline character-level model based on a token healing heuristic described in §1. While not competitive with our proposed methods, we include it here for completeness. Explain the idea, and provide pseudocode below. The experimental results are summarized in Tab. 1. We observe significantly higher surprisal (bits/byte) and error (JSD/bytes) compared to the methods in Fig. 1.

**An approximate covering inspired by token healing.** Our version of the token healing heuristic is based on an approximate covering. Given character string $\boldsymbol{\sigma} \in \Sigma^*$ and its tokenization $\delta_1 \cdots \delta_N = \tau(\boldsymbol{\sigma})$,[22] we define its **token healing approximate covering** as follows:

$$\mathcal{C}_{\text{heal}}(\boldsymbol{\sigma}) \stackrel{\text{def}}{=} \{\delta_1 \cdots \delta_{N-1} \cdot \delta' \mid \delta' \in \Delta, \kappa(\delta_N) \preceq \kappa(\delta')\} \tag{43}$$

By construction, $\kappa(\delta_1 \cdots \delta_{N-1}) \prec \boldsymbol{\sigma} \preceq \kappa(\delta_1 \cdots \delta_N)$, so it is straightforward to see that $\mathcal{C}_{\text{heal}}(\boldsymbol{\sigma}) \subseteq \mathcal{C}(\boldsymbol{\sigma})$. We also note that the size of this approximate covering is bounded by the size of the token alphabet: $|\mathcal{C}_{\text{heal}}(\boldsymbol{\sigma})| \leq |\Delta|$.

**Implementation.** The code below implements a method for efficiently constructing $\mathcal{C}_{\text{heal}}(\boldsymbol{\sigma})$. It is based on the *bundled* implementation given in App. D.

```
108  def token_healing_cover(σ):
109    if σ = ε: # no tokens to heal
110      return [Bundle(1, ε, ε, build_trie(ε)]
111    else
112      δ₁⋯δ_N ← τ(σ)
113      trie ← build_trie(δ₁⋯δ_{N-1})
114      σ' ← κ(δ_N)
115      return [Bundle(p⃗_Δ(δ₁⋯δ_{N-1}) · trie.p(σ' | ε),  δ₁⋯δ_{N-1},  σ',  trie)]
```

We first tokenize the input character sequence $\boldsymbol{\sigma}$ into a sequence of tokens $\boldsymbol{\delta}$. We then construct a probability trie for $\delta_1 \cdots \delta_{N-1}$ (i.e., all but the last token), allowing us to compute the probability of each possible next character $\boldsymbol{\sigma}'$ by querying the trie with the character suffix $\boldsymbol{\sigma}'$ corresponding to the final token.

Compared to the beam algorithm, `token_healing_cover` method represents an extreme form of pruning: it returns a beam of exactly one bundle. It also differs from the variant of token healing we described in §1, which produces token-level predictions rather than character-level predictions.

To estimate $\overrightarrow{p_\Sigma}(\cdot \mid \boldsymbol{\sigma})$ we use `next_character_probability(σ, token_healing_cover)`. However, this approximation still fails to place an appropriately high probability on `d` when conditioned on `Hello,_worl`). This happens for the same reasons as we described in §1.

**Experimental results.** The table below presents experimental results that may be compared to those of other methods in the paper. We found that the method performs poorly enough that we do not consider it a competitive approximation.

| Model | Surprisal (bits/byte) | Error (JSD/byte) | Speed (bytes/sec) |
|---|---|---|---|
| DeepSeek-R1-Distill-Llama-8B | 2.529 (2.391, 2.670) | 0.1907 (0.1800, 0.2015) | 36.70 (36.64, 36.76) |
| Llama-3.2-1B | 2.126 (2.010, 2.246) | 0.1790 (0.1691, 0.1891) | 119.35 (118.73, 119.84) |
| Meta-Llama-3.1-8B | 2.116 (1.992, 2.243) | 0.1840 (0.1738, 0.1946) | 36.61 (36.55, 36.67) |
| phi-4 (14B) | 2.204 (2.075, 2.333) | 0.1857 (0.1757, 0.1960) | 21.24 (21.13, 21.30) |

Table 1: Surprisal, JSD, and speed for the token-healing baseline across models using the same experimental settings as Fig. 1.

---

[22]If $\boldsymbol{\sigma} = \varepsilon$, $\mathcal{C}_{\text{heal}}(\varepsilon) \stackrel{\text{def}}{=} \Delta$.

# G. Probability-Based Pruning Heuristic

This section presents some additional experimental results with a more extensive pruning method, based on the relative contribution of a token string to the current cover approximation.

**Probability-based pruning heuristic.** The method works by augmenting the beam pseudocode with some additional pruning based on a threshold $\theta \in [0, 1]$. The benefit of this pruning is that it can avoid calls to bundle.extend(), which requires evaluating the language model (i.e., the performance bottleneck of our algorithm). It is often the case that for a given character string prefix, there is no reason to perform any **extend** operations because all tokenizations of that string are low probability; thus, filtering operations are needed.

```
116  def beam_{K,θ}(σ_1 ⋯ σ_N):
117    if N = 0: return [Bundle(1, ε, ε, build_trie(ε))]
118    candidates ← []
119    B ← beam(σ_1 ⋯ σ_{N−1})
120    for bundle in B:
121      filtered_bundle ← bundle.filter(σ_N)
122      if filtered_bundle is not None:
123        candidates.append(filtered_bundle)
124    τ ← θ · sum(bundle.p for bundle in candidates)
125    for bundle in B:
126      if bundle.p ≥ τ:
127        for extended_bundle in bundle.extend():
128          candidates.append(extended_bundle.filter(σ_N))
129    # Keep top-K bundles according to their prefix probability
130    return top_K(candidates, key=lambda bundle: -bundle.p)
```

**Experiments.**   Below, we extend the experiments of the main text with this additional pruning heuristic to better understand its speed and accuracy.

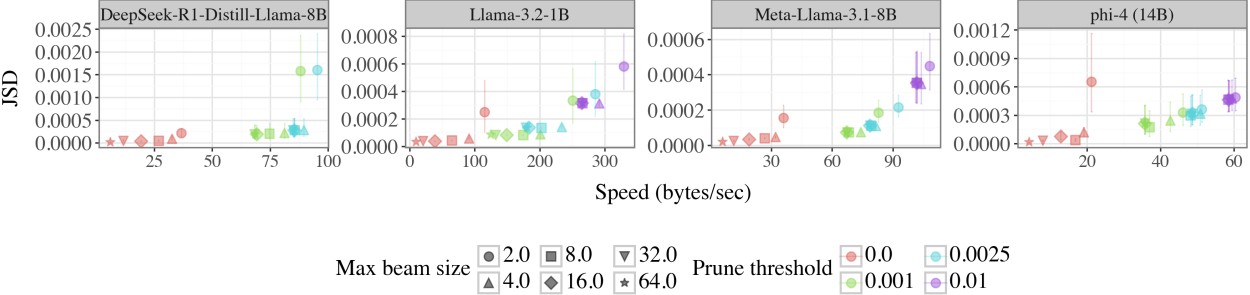

(a) Error (JSD/byte) vs. speed (bytes/sec). Here we compute the JSD between the byte-level conditional distributions computed with $\mathtt{beam}_{K,\theta}$ for $(K, \theta) \in \{4, 8, 16, 32, 64\} \times \{0, 0.001, 0.0025, 0.01\}$ and the reference model ($K = 128$, $\theta = 0$). Missing values indicate "dead-ending" (i.e., empty beam) as a result of overly aggressive pruning.

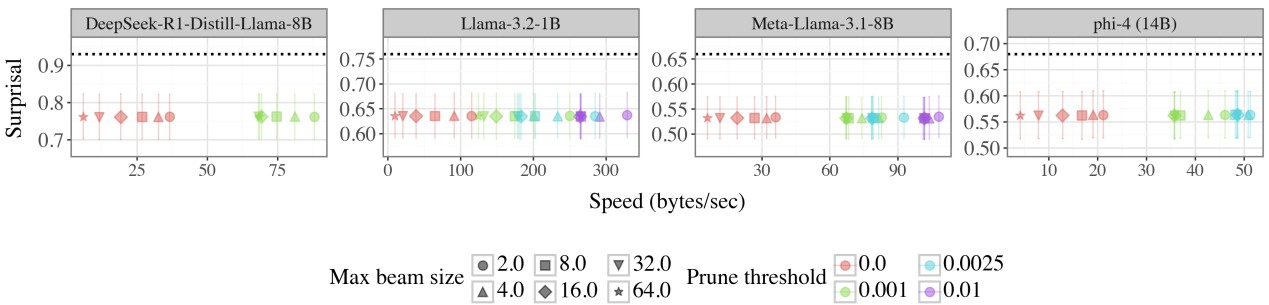

(b) Average surprisal (bits/byte) vs. speed (bytes/sec) for $\mathtt{beam}_{K,\theta}$ with $\{4, 8, 16, 32, 64\} \times \{0, 0.001, 0.0025, 0.01\}$. The dotted line represents surprisal of the canonically tokenized under the token-level language model. Missing values indicate infinite surprisal as a result of excessive pruning.

Figure 2: Additional experimental results showing the effects of the pruning thresholds on error vs. speed and surprisal vs. speed using the same settings as in §4.

