# OpenReview forum: "From Language Models over Tokens to Language Models over Characters"
_ICML.cc/2025/Conference — ICML 2025 spotlightposter_

### Official Review · Reviewer_9E2G · 2025-03-06

**Overall Recommendation:** 3

**Summary:**

The paper presents algorithms to convert token-level language models to character-level ones, addressing the prompt boundary problem. It introduces the concept of covering and proposes both exact and approximate algorithms. The main findings include that the method can accurately approximate the character-level distribution with a small computation budget. The key algorithmic idea is to find a covering of token strings that form a key technical concept, allowing the selection of members in proportion to their normalized prefix probability. The paper also provides an efficient algorithm for conditional token generation.

**Claims And Evidence:**

The claims made in the submission are supported by clear and convincing evidence. The authors provide a detailed analysis of the prompt boundary problem and demonstrate how their method resolves it. They also present empirical results on two publicly available language models, GPT2-large and Llama 3.1 8B, showing the accuracy and efficiency of their approach. The experiments include measuring the Jensen-Shannon distance (JSD) between the character-level conditional distributions with different beam sizes and the reference model, as well as the processing speed in characters per second.

**Essential References Not Discussed:**

Since I am not familiar with the relevant literature, I cannot be sure.

**Experimental Designs Or Analyses:**

The experimental designs and analyses seem valid. The authors use standard benchmark datasets (wikitext-103-v1) and modern language models (GPT2-large and Llama 3.1 8B) for evaluation. The experiments are designed to measure the accuracy and efficiency of the proposed method, and the results are presented in a clear and understandable manner. The use of beam search with different beam sizes to approximate the covering is a reasonable approach, and the trade-off between error (JSD) and speed (characters/sec) is well-documented.

**Methods And Evaluation Criteria:**

The proposed methods and evaluation criteria make sense for the problem at hand. The authors address the fundamental tension between token-level models and character-level prompts, which is a significant challenge for users of large language models. The evaluation criteria, including the JSD and processing speed, are appropriate for assessing the quality and efficiency of the proposed algorithms.

**Other Comments Or Suggestions:**

The paper is well-written and clear overall. However, there are a few minor issues. For example, in the section on "Key Properties of κ," the definition of strict-prefix monotonicity could be made more explicit with additional examples. Additionally, the notation in some parts of the paper is quite dense and could be simplified for better readability.

**Other Strengths And Weaknesses:**

The paper has several strengths. It addresses a fundamental problem in the interface between token-level language models and character-level prompts, which is a significant challenge for users of large language models. The proposed algorithms are novel and provide a principled solution to the prompt boundary problem. The empirical evaluation is thorough and demonstrates the effectiveness of the proposed methods.

However, there are also some weaknesses. The paper assumes that the language model's probability mass is concentrated around a limited set of tokenizations, which may not always be the case. Additionally, the beam summing method requires a very large beam size K if the language model does not favor a small number of tokenizations, which could be a limitation in practice.

**Questions For Authors:**

I have a few important questions for the authors:

    1. How does the proposed method handle cases where the language model's probability mass is not concentrated around a limited set of tokenizations?
    2. What are the computational requirements for the beam summing method when dealing with very large beam sizes?
    3. Could the covering concept be extended to handle more complex tokenization schemes, such as those used in multilingual language models?

**Relation To Broader Scientific Literature:**

The key contributions of the paper are related to the broader scientific literature in several ways. The work builds upon previous research on tokenization models and their limitations, such as the issues discussed by Cao & Rimell (2021) and Chirkova et al. (2023). The concept of covering is a novel contribution that addresses the prompt boundary problem, which has been highlighted in prior work like Lundberg & Ribeiro (2023). The paper also contributes to the field of computational psycholinguistics by providing a method to compute contextual surprisal for arbitrary character strings, as discussed by Giulianelli et al. (2024). The proposed algorithms and methods are likely to influence future research on language model interfaces and tokenization strategies.

**Theoretical Claims:**

I did not check the correctness of the proofs for theoretical claims in detail, but the provided proofs in the paper seem logically sound and follow standard mathematical reasoning. The proofs for Proposition 1 and Proposition 2 are based on manipulating summations and leveraging the properties of strict-prefix monotonicity, which are key concepts in the paper.

---

> ### Author Rebuttal · Authors · 2025-04-01
>
> ### General response
>
> Many reviewers suggested that our evaluation methodology, which uses a large beam as a proxy for ground truth character-level probabilities, may have some systematic bias. We will add discussion to the paper about the challenge of designing a faithful evaluation as well as the possible pitfalls in our large-beam evaluation scheme.
>
> That being said, we agree that our evaluation can be improved. In subsequent revisions of the paper, we will seek to address this in the following ways:
>
> - **Additional baselines and comparisons**:
>   - **Token healing for character-level probabilities**: Thanks to a suggestion from reviewer DnV9, we will add a comparison between our algorithm and an algorithm based on token healing for inferring character-level probabilities.
>
>   - **Perplexity per byte**: Based on the suggestion by reviewer DnV9, we will add a comparison between our method’s byte-level cross-entropy and the byte-normalized cross-entropy of a token-level language model.
>
>   - **Increased beam-width**: We will further increase the beam width of the baseline model, bringing it even closer to the ground truth and reducing any potential bias in the comparison.
>
> - **Evaluation of downstream accuracy on LM reasoning benchmarks**: We agree that some evaluation of the downstream effects of our approach would strengthen the paper.  Therefore, we will investigate the feasibility of adding an evaluation of the downstream accuracy on one or more common LM reasoning benchmarks (e.g., HellaSWAG or GLUE).  However, we do not necessarily expect to see benefits.  It's possible that randomly shifting the prompt boundary to the left would be interesting.  Consider an example from HellaSWAG:
>
>   ```
>   Prompt: "A man is sitting on a roof. He "
>   . "is using wrap to wrap a pair of skis.",
>   2. "is ripping level tiles off.",
>   3. "is holding a rubik's cube.",
>   4. "starts pulling up roofing on a roof."
>   ```
>   We would randomly shift the prompt left, e.g., move `"of. He"` from the end of the prompt to the possible continuations:
>
>   ```
>   Prompt: "A man is sitting on a ro"
>   1. "of. He is using wrap to wrap a pair of skis.",
>   2. "of. He is ripping level tiles off.",
>   3. "of. He is holding a rubik's cube.",
>   4. "of. He starts pulling up roofing on a roof."
>   ```
>
> We note, however, that we do not claim that our method must improve performance on downstream tasks, e.g., math reasoning. The purpose of our approach is to enable a character-level interface to a tokenized LM, which naturally solves the prompt boundary problem.
>
> ### Reviewer 9E2G
>
> Thank you for your review, particularly the questions!
>
> > Other Comments Or Suggestions: The paper is well-written and clear overall. However, there are a few minor issues. For example, in the section on "Key Properties of κ," the definition of strict-prefix monotonicity could be made more explicit with additional examples. Additionally, the notation in some parts of the paper is quite dense and could be simplified for better readability.
>
> Thank you for the suggestions.  We will do some editing for improving readability including examples like this will help us do that.
>
>
> > Questions For Authors:
>
> > 1. How does the proposed method handle cases where the language model's probability mass is not concentrated around a limited set of tokenizations?
>
> This is an excellent question.  Unfortunately, if mass is not concentrated, then our beam summing estimate will be skewed.  Fortunately, as the method is based on beam search, even if the beam size K is smaller than what is needed to cover the entire distribution, our method can still pick up on the more likely tokenizations.  We will add some discussion about this in the revised paper.
>
> > 2. What are the computational requirements for the beam summing method when dealing with very large beam sizes?
>
> We provide time and space complexity analyses in section 3.2.  The short answer is that time and space are linear in the beam size K.  The primary bottleneck in using more samples is that we make up to K language model calls at each position of the character string.
>
> > 3. Could the covering concept be extended to handle more complex tokenization schemes, such as those used in multilingual language models?
>
> This sounds very interesting.  What are some of the complexities that multi-lingual language models have?  If you could please provide some pointers/citations, we would love to dig into your question.

---

### Official Review · Reviewer_DnV9 · 2025-03-14

**Overall Recommendation:** 3

**Summary:**

The paper highlights the fact that language models over tokens are _not_ language models over characters, at least the way they are normally used. To be specific, the standard procedure of taking a prompt, tokenizing it, and then sampling from the model conditioned on the token sequences is _not_ the same thing as sampling from the model's distribution conditioned on the prompt as a textual prefix. The paper introduces the idea of a _covering_ which is a sufficient set of token sequences that, if evaluated by the model, allow one to sample properly from the text conditioned distribution. An approximation of this covering is also given for computational convenience.

**Claims And Evidence:**

The paper proceeds very logically from its premise to conclusions. I believe all of the theoretical claims made are well-supported.

**Essential References Not Discussed:**

I'm not aware of any prior papers on this exact topic. All of the related papers I know of are discussed fairly in the paper.

**Experimental Designs Or Analyses:**

The existing experiments are valid and I appreciate the results. But I think the paper misses a chance to really motivate the problem. Basically the existing comparison is between the proposed algorithm with a small beam width and the same algorithm with a large beam width. I think the paper would be much stronger if a comparison was made between the proposed algorithm and the existing practice of simply tokenizing the prefix and sampling as well as the token healing correction heuristic. Of course, neither of these are the "correct" thing and so I would expect the proposed algorithm to beat them soundly, especially with a large beam width. If there is a big difference, then there's a clear motivation for using the proposed algorithm if one cares about byte conditioned sampling. On the other hand, if there isn't a big difference, it means that the heuristics may be "good enough" which is something we didn't know before!

**Methods And Evaluation Criteria:**

I think the method proposed (using the full cover) is exactly the correct thing to do, although it is not very practical due to its exponential time complexity (which is unavoidable). The approximation given is the most natural approximation of the full problem.

The evaluation criterion (JSD on next byte) is valid but I do think leaves something to be desired. I would be interested in seeing the cross-entropy loss at the byte level, this could be compared to the byte normalized loss of the LM at the token level (i.e. the "bits per byte") which is obtained by converting the standard cross entropy loss from units of nats/token to bits/byte (for a certain text).

**Other Comments Or Suggestions:**

I think the footnote corresponding to "10" in Figure 1 seems to be missing?

**Other Strengths And Weaknesses:**

I really appreciate the clarity of the presentation. I found the paper very enjoyable to read.

**Questions For Authors:**

Some of the existing works on the marginalization problem (e.g. Cao & Rimell, 2021; Chirkova et al., 2023) tackle the problem using importance sampling, which has the benefit of being unbiased. Is there any way to apply a similar idea in the setting of conditional generation?

**Relation To Broader Scientific Literature:**

This paper can be thought of as an extension of the work of marginalizing over segmentations of a text (Cao & Rimell, 2021; Chirkova et al., 2023) but done in a manner that is "open to the right," which allows one to consider sampling an extension of the text.

**Theoretical Claims:**

I did not carefully check the proofs in the appendix, but I strongly believe the propositions are correct.

---

> ### Author Rebuttal · Authors · 2025-04-01
>
> ### General response
>
> Many reviewers suggested that our evaluation methodology, which uses a large beam as a proxy for ground truth character-level probabilities, may have some systematic bias. We will add discussion to the paper about the challenge of designing a faithful evaluation as well as the possible pitfalls in our large-beam evaluation scheme.
>
> That being said, we agree that our evaluation can be improved. In subsequent revisions of the paper, we will seek to address this in the following ways:
>
> - **Additional baselines and comparisons**:
>   - **Token healing for character-level probabilities**: Thanks to a suggestion from reviewer DnV9, we will add a comparison between our algorithm and an algorithm based on token healing for inferring character-level probabilities.
>
>   - **Perplexity per byte**: Based on the suggestion by reviewer DnV9, we will add a comparison between our method’s byte-level cross-entropy and the byte-normalized cross-entropy of a token-level language model.
>
>   - **Increased beam-width**: We will further increase the beam width of the baseline model, bringing it even closer to the ground truth and reducing any potential bias in the comparison.
>
> - **Evaluation of downstream accuracy on LM reasoning benchmarks**: We agree that some evaluation of the downstream effects of our approach would strengthen the paper.  Therefore, we will investigate the feasibility of adding an evaluation of the downstream accuracy on one or more common LM reasoning benchmarks (e.g., HellaSWAG or GLUE).  However, we do not necessarily expect to see benefits.  It's possible that randomly shifting the prompt boundary to the left would be interesting.  Consider an example from HellaSWAG:
>
>   ```
>   Prompt: "A man is sitting on a roof. He "
>   . "is using wrap to wrap a pair of skis.",
>   2. "is ripping level tiles off.",
>   3. "is holding a rubik's cube.",
>   4. "starts pulling up roofing on a roof."
>   ```
>   We would randomly shift the prompt left, e.g., move `"of. He"` from the end of the prompt to the possible continuations:
>
>   ```
>   Prompt: "A man is sitting on a ro"
>   1. "of. He is using wrap to wrap a pair of skis.",
>   2. "of. He is ripping level tiles off.",
>   3. "of. He is holding a rubik's cube.",
>   4. "of. He starts pulling up roofing on a roof."
>   ```
>
> We note, however, that we do not claim that our method must improve performance on downstream tasks, e.g., math reasoning. The purpose of our approach is to enable a character-level interface to a tokenized LM, which naturally solves the prompt boundary problem.
>
>
> ### Response to Reviewer DnV9
>
> Thank you so much for your review. It has been incredibly helpful as your suggestions for additional baselines are fantastic, and we will add them (more below).
>
> > re: "bits per byte" suggestion
>
> Thank you for this suggestion! This is a great comparison to run, and we plan to add it to our experimental evaluation.
>
> > re: a token-healing-based baseline
>
> This is an incredibly good suggestion. Coincidentally, we used that method precisely during debugging but didn't think of adding it as a baseline.  What an excellent idea - thank you!
>
> > I think the footnote corresponding to "10" in Figure 1 seems to be missing?
>
> It will be fixed in the next revision.  The text for the two footnotes in Fig 1 (i.e., 10 and 11) appear on the previous page, on lines 376–384 [left column], as footnotes labeled with the same numbers.
>
> > Re: Importance sampling
>
> Importance sampling could be used to estimate the prefix probability (and indeed it is unbiased for that).  However, it does not give an unbiased conditional prefix probability, as it requires the division of two prefix probabilities.  We briefly experimented with sequential Monte Carlo but found that noncanonical token sequences were strongly overrepresented.  For example, when we use it on the string `"SELECT * FROM"` roughly 94% of GPT-2's samples start with the token `S` rather than the canonical token `SELECT`. So, we ended up ruling it out in favor of beam search.  Perhaps a more sophisticated proposal distribution would help.  We suspect that it is useful to represent particles as "buckets" as we did in our pruning heuristic (starting on line 337).
>
> It's possible that we should add something about this 'negative result' to an appendix.

---

> > ### Comment · Reviewer_DnV9 · 2025-04-05
> >
> > I would like to thank the authors for their rebuttal. I think with the proposed changes the work will be much more solid, so I have increased my score.

---

### Official Review · Reviewer_zQVq · 2025-03-15

**Overall Recommendation:** 4

**Summary:**

This paper presents an algorithm for converting token-level language models for character level language models. The authors present compelling analysis as well as detailed explanation. This work also includes practical experimental evaluation results.

**Claims And Evidence:**

The claims are backed by convincing theoretical and empirical analysis.

**Essential References Not Discussed:**

N/A.

**Experimental Designs Or Analyses:**

See my concerns in Methods And Evaluation Criteria. The experimental design and analysis is largely convincing.

**Methods And Evaluation Criteria:**

The authors evaluated their algorithms using GPT2 and Llama 8B models on the Wikitext dataset—a reasonable choice. They measured performance using Jensen-Shannon distance between a high-budget model (treated as ground truth) and various faster approximation models. While this approach makes practical sense, it remains somewhat unsatisfying. Could the authors compare their methods against a truly exact model using full coverings, at least in small-scale settings? This would provide a more rigorous baseline for evaluation.

**Other Comments Or Suggestions:**

N/A.

**Other Strengths And Weaknesses:**

Strength:

- Generally the paper is well written, the authors made considerable effort to put together this research.
- The research topic and idea is very novel and compelling.
- I really appreciate the author's effort in systematically explaining the problem, existing solutions and their shortcomings. The explanations are valuable contributions in and of itself.

Weakness:

- Would be great if the authors can repeat the comparison with an exact character model.

**Questions For Authors:**

N/A.

**Relation To Broader Scientific Literature:**

To the best of my knowledge, this paper is very upfront about introducing prior work, including mentioning existing solution to the token boundary problem using token healing heuristics, and also presents compelling analysis revealing their short comings.

**Theoretical Claims:**

They appear reasonable.

---

> ### Author Rebuttal · Authors · 2025-04-01
>
> ### General response
>
> Many reviewers suggested that our evaluation methodology, which uses a large beam as a proxy for ground truth character-level probabilities, may have some systematic bias. We will add discussion to the paper about the challenge of designing a faithful evaluation as well as the possible pitfalls in our large-beam evaluation scheme.
>
> That being said, we agree that our evaluation can be improved. In subsequent revisions of the paper, we will seek to address this in the following ways:
>
> - **Additional baselines and comparisons**:
>   - **Token healing for character-level probabilities**: Thanks to a suggestion from reviewer DnV9, we will add a comparison between our algorithm and an algorithm based on token healing for inferring character-level probabilities.
>
>   - **Perplexity per byte**: Based on the suggestion by reviewer DnV9, we will add a comparison between our method’s byte-level cross-entropy and the byte-normalized cross-entropy of a token-level language model.
>
>   - **Increased beam-width**: We will further increase the beam width of the baseline model, bringing it even closer to the ground truth and reducing any potential bias in the comparison.
>
> - **Evaluation of downstream accuracy on LM reasoning benchmarks**: We agree that some evaluation of the downstream effects of our approach would strengthen the paper.  Therefore, we will investigate the feasibility of adding an evaluation of the downstream accuracy on one or more common LM reasoning benchmarks (e.g., HellaSWAG or GLUE).  However, we do not necessarily expect to see benefits.  It's possible that randomly shifting the prompt boundary to the left would be interesting.  Consider an example from HellaSWAG:
>
>   ```
>   Prompt: "A man is sitting on a roof. He "
>   . "is using wrap to wrap a pair of skis.",
>   2. "is ripping level tiles off.",
>   3. "is holding a rubik's cube.",
>   4. "starts pulling up roofing on a roof."
>   ```
>   We would randomly shift the prompt left, e.g., move `"of. He"` from the end of the prompt to the possible continuations:
>
>   ```
>   Prompt: "A man is sitting on a ro"
>   1. "of. He is using wrap to wrap a pair of skis.",
>   2. "of. He is ripping level tiles off.",
>   3. "of. He is holding a rubik's cube.",
>   4. "of. He starts pulling up roofing on a roof."
>   ```
>
> We note, however, that we do not claim that our method must improve performance on downstream tasks, e.g., math reasoning. The purpose of our approach is to enable a character-level interface to a tokenized LM, which naturally solves the prompt boundary problem.
>
> ### Response to Reviewer zQVq
>
> Thank you for the thoughtful review and the kind words.
>
> > Methods And Evaluation Criteria: The authors evaluated their algorithms using GPT2 and Llama 8B models on the Wikitext dataset—a reasonable choice. They measured performance using Jensen-Shannon distance between a high-budget model (treated as ground truth) and various faster approximation models. While this approach makes practical sense, it remains somewhat unsatisfying. Could the authors compare their methods against a truly exact model using full coverings, at least in small-scale settings? This would provide a more rigorous baseline for evaluation.
>
> Unfortunately, using a truly exact model is not feasible as marginalizing over tokenization of even short strings quickly becomes infeasible.  We don't think that short string comparisons of beam summing's performance would necessarily generalize to the case of longer strings.  So we worry that it might be misleading.
>
> Another option we considered was evaluating our method's ability to convert a probabilistic context-free grammar over a tokenized set of symbols into a character-level model, as it is possible to compute those probabilities in cubic time; however, it is unclear whether the results would generalize to the general LM setting.  We are happy to discuss this option further in the discussion period.

---

### Official Review · Reviewer_us9y · 2025-03-19

**Overall Recommendation:** 3

**Summary:**

This paper is motivated by addressing the "prompt boundary problem" in token-level language models. In models using tokenizers like BPE, even small changes at the prompt boundary, such as adding a whitespace, can dramatically alter the next token distribution in unintuitive ways, which is undesired behavior.
The authors propose a principled solution to convert token-level language models into character-level ones. Their method is built around the concept of a "covering" - the set of all minimal token sequences that, when decoded, would produce a given character string or a string having it as a prefix. By considering the weighted probability distribution across this entire covering rather than just a single tokenization, the method correctly computes character-level probabilities from token-level language models.
The paper presents both exact algorithms and efficient approximations based on beam search for computing these character-level probabilities. Their experiments show that with reasonable computational resources, their method achieves high accuracy in estimating character-level distributions with minimal error at practical speeds on models like Llama 3.1 8B, effectively solving the prompt boundary problem and creating a more intuitive interface for working with tokenized language models.

**Claims And Evidence:**

yes

**Essential References Not Discussed:**

no

**Experimental Designs Or Analyses:**

please see above

**Methods And Evaluation Criteria:**

For the experiment, the ground truth in this evaluation is an approximation obtained using a very high beam size (K = 128) from the same method, it may inherit the biases or errors of that approximation. Evaluating on real-world datasets would provide stronger evidence that the method effectively resolves the prompt boundary problem in practical scenarios, rather than only on controlled benchmarks.

**Other Comments Or Suggestions:**

no

**Other Strengths And Weaknesses:**

Pros:
1. The paper effectively formalizes the challenge of converting token-level LLMs to express character-level outputs. Its formulation is clear and the use of color-coded notations enhances readability.
2. The proposed method incorporates beam search pruning to significantly improve efficiency.
3. The authors demonstrate its effectiveness under a reasonable computational budget.

Cons:
1. The evaluation is primarily conducted on a single corpus. Expanding experiments to include diverse, real-world datasets would help validate the method’s robustness and provide more practical insights into how it improves actual problem-solving.
2. The evaluation is evaluated against the same method but with higher beam search K, this may introduce bias.

**Questions For Authors:**

please see other sec

**Relation To Broader Scientific Literature:**

This paper is related to tokenization in LLMs, prior works have identified the problems with token-level models processing character-level prompts, this paper formalizes this problem of prompt boundary. This work also relates to constrained decoding works, where converting a token-level LLM to a character level LLM helps in constrained decoding.

**Theoretical Claims:**

yes

---

> ### Author Rebuttal · Authors · 2025-04-01
>
> ### General response
>
> Many reviewers suggested that our evaluation methodology, which uses a large beam as a proxy for ground truth character-level probabilities, may have some systematic bias. We will add discussion to the paper about the challenge of designing a faithful evaluation as well as the possible pitfalls in our large-beam evaluation scheme.
>
> That being said, we agree that our evaluation can be improved. In subsequent revisions of the paper, we will seek to address this in the following ways:
>
> - **Additional baselines and comparisons**:
>   - **Token healing for character-level probabilities**: Thanks to a suggestion from reviewer DnV9, we will add a comparison between our algorithm and an algorithm based on token healing for inferring character-level probabilities.
>
>   - **Perplexity per byte**: Based on the suggestion by reviewer DnV9, we will add a comparison between our method’s byte-level cross-entropy and the byte-normalized cross-entropy of a token-level language model.
>
>   - **Increased beam-width**: We will further increase the beam width of the baseline model, bringing it even closer to the ground truth and reducing any potential bias in the comparison.
>
> - **Evaluation of downstream accuracy on LM reasoning benchmarks**: We agree that some evaluation of the downstream effects of our approach would strengthen the paper.  Therefore, we will investigate the feasibility of adding an evaluation of the downstream accuracy on one or more common LM reasoning benchmarks (e.g., HellaSWAG or GLUE).  However, we do not necessarily expect to see benefits.  It's possible that randomly shifting the prompt boundary to the left would be interesting.  Consider an example from HellaSWAG:
>
>   ```
>   Prompt: "A man is sitting on a roof. He "
>   . "is using wrap to wrap a pair of skis.",
>   2. "is ripping level tiles off.",
>   3. "is holding a rubik's cube.",
>   4. "starts pulling up roofing on a roof."
>   ```
>   We would randomly shift the prompt left, e.g., move `"of. He"` from the end of the prompt to the possible continuations:
>
>   ```
>   Prompt: "A man is sitting on a ro"
>   1. "of. He is using wrap to wrap a pair of skis.",
>   2. "of. He is ripping level tiles off.",
>   3. "of. He is holding a rubik's cube.",
>   4. "of. He starts pulling up roofing on a roof."
>   ```
>
> We note, however, that we do not claim that our method must improve performance on downstream tasks, e.g., math reasoning. The purpose of our approach is to enable a character-level interface to a tokenized LM, which naturally solves the prompt boundary problem.
>
> ### Response to Reviewer us9y
>
> Thank you for the review and constructive suggestions for improving our experimental evaluation.  [We are also glad you appreciated the color coding!]
>
> > For the experiment, the ground truth in this evaluation is an approximation obtained using a very high beam size (K = 128) from the same method, it may inherit the biases or errors of that approximation. Evaluating on real-world datasets would provide stronger evidence that the method effectively resolves the prompt boundary problem in practical scenarios, rather than only on controlled benchmarks.
>
> This is a reasonable concern: Essentially, the concern is that beam size K=64 might be really good at predicting K=128 because they systematically make the same kinds of errors.  We treat K=128 as essentially ground truth, which we have not proven is the case.  We will discuss this issue in the revised paper, as it is an important limitation of our experiment design.  Thank you.
>
> > The evaluation is primarily conducted on a single corpus. Expanding experiments to include diverse, real-world datasets would help validate the method’s robustness and provide more practical insights into how it improves actual problem-solving.
>
> Thanks for this suggestion! We will investigate the feasibility of adding a downstream accuracy evaluation on a common LM benchmark, e.g., HellaSWAG, GLUE.  Note that we do not intend to claim that our method will necessarily improve model performance on downstream tasks (unless there are prompt boundary issues).  We emphasize that enabling a character-level interface to language models is valuable in its own right—for instance, when measuring character-level surprise for psycholinguistic experiments ([Giulianelli et al., 2024](https://arxiv.org/abs/2410.02691)) or handling tasks that inherently require fine-grained character-level control.
>
> > The evaluation is evaluated against the same method but with higher beam search K, this may introduce bias.
>
> Great point.  Please see the general response.

---

### Decision · Program_Chairs · 2025-05-01

**Decision:**

Accept (spotlight poster)

**Comment:**

The paper investigates the problem that language models are sensitive to the specification of the prompt (e.g., if the prompt ends with a space or not). They proceed logically, derive theoretical insights and a practical algorithm that seems to perform well. This meta-reviewer is particularly excited because the paper studies an important problem from an application point of view and provides a solution which is both theoretically and empirically sound.